# The Unreasonable Effectiveness of Scaling Agents for Computer Use

## Abstract

Computer use agents (CUAs) hold promise for automating everyday digital tasks, but their unreliability and high variance hinder their application to long-horizon, complex tasks. We introduce *Behavior Best-of-N (bBoN)*, a method that scales over agents by generating multiple rollouts and selecting among them using behavior narratives that describe the agents' rollouts. It enables both wide exploration and principled trajectory selection, substantially improving robustness and success rates. On OSWorld, our bBoN scaling method establishes a new state of the art (SoTA) at 69.9%, significantly outperforming prior methods and approaching human-level performance at 72%, with comprehensive ablations validating key design choices. We further demonstrate strong generalization results to different operating systems on WindowsAgentArena and AndroidWorld. Crucially, our results highlight the unreasonable effectiveness of scaling CUAs, when you do it right: effective scaling requires structured trajectory understanding and selection, and bBoN provides a practical framework to achieve this.

## 1 Introduction

Computer-use agents (CUAs) offer the promise of automating everyday digital tasks across operating systems and applications (Xie et al., 2024; Song et al., 2025; Guo et al., 2025b; Yang et al., 2025b; Xie et al., 2025b; Wang et al., 2025b;c). Yet despite rapid advances, current CUAs remain unreliable on long-horizon, complex problems. The difficulty lies not only in solving individual steps but also in sustaining correctness across dozens or even hundreds of inter-actions. Small mistakes accumulate, feedback is often delayed, so-lution paths branch in unpredictable ways, and environmental noise (UI changes, pop-ups, latency) further destabilizes performance (Yang et al., 2025a). Together, these factors cause high variance in out-comes: the same agent may succeed on one attempt but fail catastrophically on another.

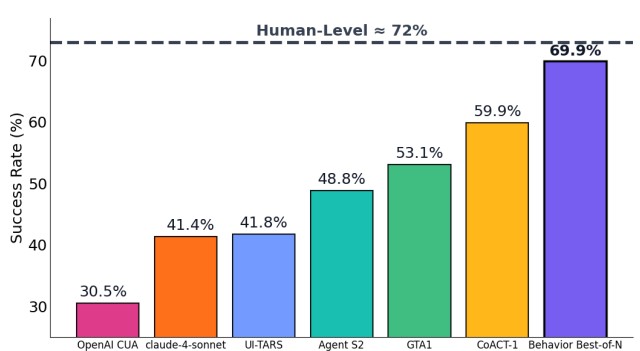

Figure 1: Performance on OSWorld at 100 steps. Our method beats the previous SoTA by 10% absolute improve-ment, nearly reaching human level performance.

A natural way to mitigate this fragility is *wide scaling*: instead of simply accepting a single rollout from one agent, we can scale the number of agents to generate multiple rollouts in parallel and select the best. This wide scaling perspective leverages the fact that agents, while suboptimal individually, often succeed on complementary subsets of tasks, as shown in Figure 2. However, scaling CUAs in-troduces unique challenges. First, long-horizon trajectories are information-dense with multimodal details, most of which are irrelevant to task success, making them difficult to represent, interpret, and compare. Second, evaluation itself is non-trivial: many computer-use tasks admit multiple valid solutions, and automatic evaluation struggles to decide whether a trajectory is correct (Xie et al.,

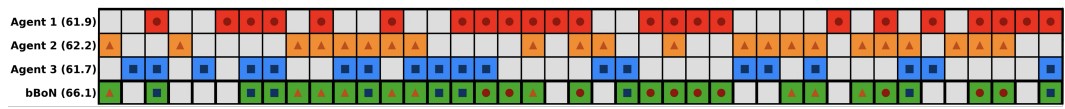

Figure 2: Disjoint task success across rollouts by three agent instances. Behavior Best-of-N (bBoN) leverages this complementarity by selecting the best trajectory among multiple rollouts.

2024; Rawles et al., 2025; Bonatti et al., 2024). Therefore, scaling CUAs effectively demands new methods for compactly representing trajectories and reliably evaluating them.

To address these challenges, we introduce Behavior Best-of-N (bBoN), a novel framework that enables wide scaling of CUAs. Our approach first converts raw trajectories into behavior narratives: concise summaries that capture what the agent actually did and how it affected the environment, preserving task-relevant action–effect summaries while filtering away irrelevant detail at individual steps. These narratives provide a compact yet faithful representation that makes it easier for a judge to compare candidates. bBoN then performs selection directly over narratives, enabling reliable selection among multiple rollouts. In addition, we build upon existing CUAs and introduce an improved baseline computer use agentic framework to generate high quality trajectories for bBoN.

Our method delivers unreasonably strong performance on computer-use benchmarks. On OS-World (Xie et al., 2024), it achieves a new state of the art with a 69.9% success rate (100 steps), surpassing the previous best of 59.9% and approaching human-level performance at 72% (Figure 1). Beyond OSWorld, our approach also demonstrates strong zero-shot generalizability on WindowsAgentArena (Bonatti et al., 2024) and AndroidWorld (Rawles et al., 2025).

Our contributions are four-fold:

- We introduce the wide scaling paradigm for CUAs, showing that generating multiple trajectories in parallel and selecting among them substantially improves robustness and coverage.
- We propose Behavior Best-of-N (bBoN), a framework that converts dense trajectories into compact behavior narratives and uses them for principled trajectory selection.
- Our method, together with an improved CUA baseline, achieves a new SoTA of 69.9% on OSWorld, surpassing prior work by a large margin (10% absolute improvement) and approaching human performance at 72%.
- We provide extensive ablations validating our design choices and demonstrate strong zero-shot generalizability on WindowsAgentArena and AndroidWorld.

## 2 BACKGROUND

### 2.1 COMPUTER-USE AGENTS

Computer-use agents (CUAs) executing user instructions can be framed as a partially observable Markov Decision Process (POMDP) defined as $\mathcal{M} = \langle \mathcal{S}, \mathcal{O}, \mathcal{A}, \mathcal{T}, \mathcal{I}, R \rangle$, where $\mathcal{S}$ is the state space encoding the computer state, $\mathcal{O}$ is the observation space such as desktop screenshots, $\mathcal{A}$ is the action space of the agent (e.g. `agent.click(...)` and `agent.type(...)`), $\mathcal{T} : \mathcal{S} \times \mathcal{A} \to \Delta(\mathcal{S})$ is a stochastic transition function, $\mathcal{I}$ is the space of possible user instructions represented in natural language, and $R : (\mathcal{S} \times \mathcal{A})^* \times \mathcal{I} \to [0, 1]$ denotes the instruction reward function that assigns a scalar reward to a trajectory of states and actions $\tau := (s_0, a_0, \ldots, a_{T-1}, s_t)$ on task $I$. We use $h_t := (o_0, a_0, \ldots, o_{t-1}, a_{t-1}, o_t)$ to denote a time-ordered history of all consecutive observations and actions up to and including $o_t$.

A broad spectrum of computer agents has been explored including general agentic frameworks (Song et al., 2025; Yang et al., 2025b; Agashe et al., 2025; 2024), generalist agents (Anthropic, 2025; OpenAI, 2024; Guo et al., 2025a) and graphical user interface (GUI) agents (Wang et al., 2025a; Xu et al., 2025). These prior work consider a single model as the policy $\pi(a|h_t, I)$ that, when executed, yields one trajectory $\tau = (o_0, a_0, \ldots, o_T)$ where $a_t \sim \pi(\cdot|h_t, I)$. In contrast, our work is the first, to our knowledge, that focuses on scaling the number of candidate solution trajectories by using multiple base models and policies, and we propose effective methods to select the optimal solution.

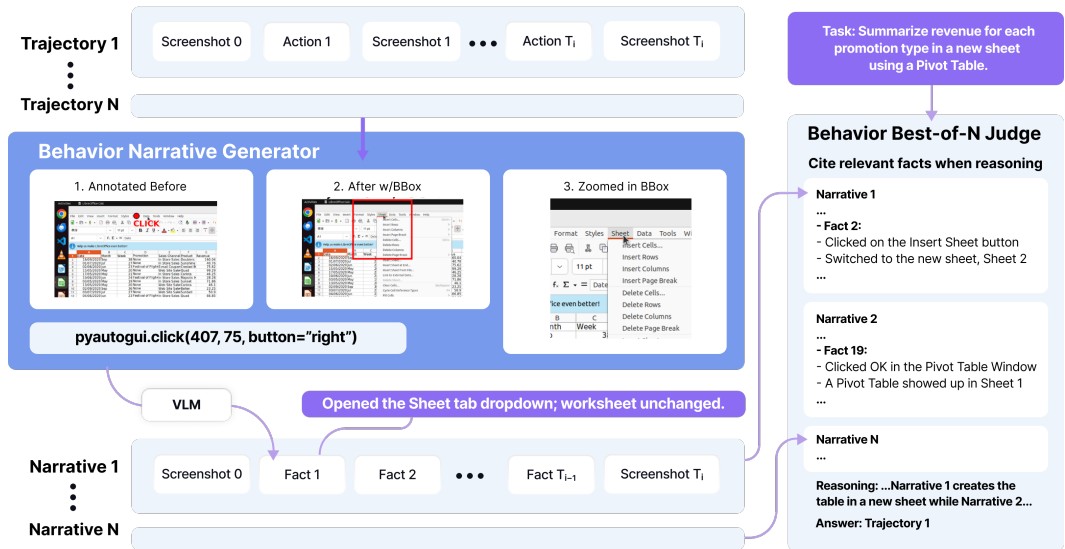

Figure 3: Behavior Best-of-N generates multiple rollouts consisting of screenshots and actions. These trajectories are converted into behavior narratives via the behavior narrative generator, using the executed action and before/after screenshots to describe what was changed. Finally, the behavior narratives are provided to the judge which selects the best trajectory through comparison.

## 2.2 TEST-TIME SCALING

A common strategy for improving large multimodal models and their agentic extensions is through test-time scaling (Zhu et al., 2025), where multiple solutions are generated either in parallel or sequentially, followed by selection a final response using a reward model or iterative generation of new solutions (Snell et al., 2024; Lightman et al., 2024). Recent work (Yang et al., 2025b) has adapted this idea for CUAs with *step-wise BoN* (Zhu et al., 2025), where at each step the agent $\pi$ generates $K$ candidate actions $\mathcal{C}_t = \{a_t^{(k)}\}_{k=1}^{K} \sim \pi(\cdot|h_t, I)$ and then a judge $J$ selects the best action $\hat{a} = J(\mathcal{C}_t)$. While this can help with local improvements, it commits the rollout to the current agent plan. In tasks with multiple valid solutions paths, this can lead the agent to over-commit to a harder route, missing easier alternatives. In contrast, our work investigates the *wide scaling* approach using trajectory-level BoN, where a final best trajectory is selected among a set of candidates trajectories generated by multiple base agents or models.

However, implementing trajectory-level BoN is non-trivial because trajectory evaluation is still a fundamental challenge. Most existing benchmarks such as OSWorld (Xie et al., 2024), WindowsAgentArena (Bonatti et al., 2024), and AndroidWorld (Rawles et al., 2025) use evaluation scripts written by humans which cannot be scaled. In contrast, work on web-agent benchmarks, a subset of CUA focused on browsers, has explored using vision-language models (VLMs) as judges (He et al., 2024; Deng et al., 2023; Xue et al., 2025). However, these judges are typically tuned for the web domain, require human-defined rubrics, and do not generalize well to the broader tasks faced by CUAs. In addition, aligning such judges with human judgment requires substantial manual effort, such as in Mind2Web 2 (Gou et al., 2025) that achieved 99% agreement using code-generated rubrics but still relied on extensive human verification. Moreover, all these evaluation methods only work with a single trajectory. Our work aims to augment trajectory-level BoN to handle trajectory evaluation by (1) improving trajectory understanding by converting trajectories into a behavior narrative that describes what an agent did and (2) comparing trajectories using the behavior narratives to effectively distinguish the best.

## 3 METHOD

Our **Behavior Best-of-N** framework, shown in Figure 3, is designed to enable wide scaling over many agent rollouts. We improve upon Agent S2 (Agashe et al., 2025), a top-performing open-

source agentic framework, and introduce two key components: *Behavior Narrative Generator* and *Behavior Best-of-N Judge*. Given a rollout, the Behavior Narrative Generator derives facts from each transition, yielding a behavior narrative that describes what the agent did (action-effects) while discarding irrelevant details. The Behavior Best-of-N Judge then conducts a comparative evaluation of the candidate narratives across multiple rollouts to determine the best solution.

### 3.1 BEHAVIOR NARRATIVE GENERATION

Long-horizon trajectories are information-dense, with every step producing a new screenshot. We argue that it is not necessary or even optimal to judge all of the raw visual content directly to understand what actually occurred. We propose to extract the task-relevant changes caused by the agent's actions from screenshots in order for a downstream judge to focus on the changes that matter. We construct a behavior narrative composed of facts that describe what the agent did at each step. Concretely, given a generator $G$ (instantiated using a VLM) and an agent rollout $\tau = (s_0, a_1, s_1, \ldots, a_{T-1}, s_T)$ where $s$ denotes a screenshot and $a$ denotes an agent action, we feed in transitions $(s_i, a_i, s_{i+1})$ to the generator and derive facts $\phi_i = G(s_i, a_i, s_{i+1})$, for each $i \in \{0, \ldots, T-1\}$.

To generate accurate facts, the Behavior Narrative Generator takes in a screenshot before action execution, the action to execute, and the screenshot after execution as depicted in Figure 3. The generator applies targeted visual augmentations for pointer interactions (clicks, taps, moves, and drags), as these actions require pixel-level precision and are more prone to agent hallucination. For example, a step-level hallucination where a click on the Save button fails but the agent believes otherwise can be the difference between a success or failure. On the screenshot before action execution $s_i$, we overlay a marker centered at the pointer coordinate $(x_i, y_i)$ where $a_i$ will occur. On the screenshot after action execution $s_{i+1}$, we extract a zoomed crop $s_{i+1}^z$ of a fixed-size square centered at the final pointer coordinate $(x_{i+1}, y_{i+1})$ and outline the crop in $s_{i+1}$ to indicate the region of interest. The zoom provides the generator with fine-grained evidence to verify that the intended change occurred. To handle cases where changes are delayed (e.g. clicking a hyperlink), screenshot $s_{i+1}$ is taken 3 seconds after action execution.

Once facts have been derived from each transition, we construct a behavior narrative $\tilde{\tau} = (s_0, \phi_0, \phi_1 \ldots, \phi_{T-1}, s_T)$ that retains only task-relevant changes. We include the initial and final screenshot to ground where changes begin from and what they result in. This allows Behavior Best-of-N to focus solely on what the agent did differently between trajectories.

### 3.2 BEHAVIOR BEST-OF-N JUDGE

While generating multiple rollouts increases the chance that at least one rollout is successful, the benefits can only be realized if we can reliably select the correct trajectory. Selection is challenging because a judge must both interpret long-horizon behavior within each rollout (to verify task requirements) and discriminate among candidates. To simplify this, we decide to separate these responsibilities by generating a concise behavior narrative that describes the long-horizon behavior so the bulk of the judge's responsibility lies on selecting between candidates. We therefore apply Behavior Best-of-N (bBoN) to the behavior narratives $\tilde{\tau}$ produced through behavior narrative generation, so the judge can focus on differences between agent behaviors.

Concretely, given a set of base policies $\{\pi_m\}_{m=1}^{M}$, we generate candidates $\mathcal{C} = \bigcup_{m=1}^{M} \{\tau_m^{(n)}\}_{n=1}^{N_m}$ where each candidate $\tau_m^{(n)}$ is sampled via stochastic decoding from a base policy $\pi_m$. This allows us to capture diversity from variance within the same model ($n = 1 \ldots N_m$) and differing capabilities across different models ($m = 1 \ldots M$). Our objective is to select the candidate trajectory that maximizes task return $\hat{\tau} \in \arg\max_{\tau \in \mathcal{C}} R(\tau, I)$. The candidate set $\mathcal{C}$ is converted to a corresponding set of behavior narrative candidates $\tilde{\mathcal{C}} := \{\tilde{\tau}^{(n)}\}_{n=1}^{|C|}$, according to the behavior narrative generation in Section 3.1. Then a VLM judge $J$ is prompted to run comparative evaluation using all narratives in $\tilde{\mathcal{C}}$ and select a single best narrative candidate, which corresponds to the final selected trajectory $\hat{\tau} \in \mathcal{C}$. In this work, we instantiate comparative evaluation using a single-round multiple-choice question (MCQ) format, which enables a more informed comparison than independent ranking while being more token-efficient and faster than multi-round tournament-style comparisons of subsets of candidates. The system prompt (Section H) emphasizes on citing and contrasting facts to ensure each

candidates' behaviors are carefully observed, which we find gives small improvements (Section F). By comparing behavior narratives altogether, we enable wide scaling over many agents.

### 3.3 AN IMPROVED AGENTIC FRAMEWORK BASELINE

As Behavior Best-of-N operates on multiple full-length trajectories generated by base agents, we can improve the overall performance and latency of bBoN by starting with the best frameworks for the base agents. Inspired by Agashe et al. (2025) and Song et al. (2025), we created an improved baseline agentic framework, *Agent S3*, which achieves a new SoTA even before incorporation into bBoN. It draws upon two key ideas: 1) performance gains of programmatic edits over direct GUI manipulation when needed (up to the agent itself), and 2) speedup by using a flat (worker only) policy instead of a manager-worker hierarchy.

**Coding Agent**    To encourage diverse solution paths, our GUI policy $\pi(a_t \mid I, h_t)$ reasons what approach might be best suited for the next step: generate a GUI action $a_t \in \mathcal{A}_{\mathrm{gui}}$ or invoke the *coding agent* for programmatic edits (e.g., bulk operations, file transforms, structured parsing). A code call launches a bounded inner loop with budget $B$ that iterates on generated code and terminal *feedback*. At inner step $k$, the coding agent conditions on $c_k^{\mathrm{code}} = (I, o_t, F_{1:k-1})$, where $F_{1:k-1}$ aggregates execution signals (status, return code, stdout/stderr) from prior iterations. It either emits Python/Bash to be executed in a sandboxed VM, or returns a control token DONE/FAIL. On termination, a brief summary of the session—logic, observed effects, and a verifiable inspection checklist—is appended to the GUI agent's history to aid on-screen verification and subsequent planning by the GUI policy. Different from Song et al. (2025), our coding agent implementation does not use the AutoGen Wu et al. (2023) framework nor does it use an orchestrator to divide and delegate tasks across the GUI and coding agents. Our coding agent implementation is natively integrated into our GUI agent's action space, allowing GUI agent to reason when best to delegate the next step to the coding agent.

**Flat Policy**    We remove hierarchical planning in favor of a flat policy that can replan at any time based on $(I, h_t)$. Contemporary foundation models exhibit strong GUI understanding and can maintain short-horizon plans in context, making a separate high-level planner unnecessary and sometimes counterproductive (e.g., when subgoals become stale). We evaluate these design choices in Table 2; implementation details appear in Section D.

## 4 EXPERIMENTS AND ANALYSIS

In the following experiments, we systematically investigate the effectiveness of Behavior Best-of-N (bBoN) across several dimensions of computer-use agents. Specifically, we aim to address the following research questions:

1) *Performance.* How does bBoN perform compared with other CUA baselines?

2) *Scalability.* How does performance scale with increasing number of rollouts?

3) *Ensembling.* How should we select a mixture-of-models ensemble?

4) *Representation.* How do behavior narratives compare to other trajectory representations?

5) *Selection mechanism.* How does comparative selection compare to independent ranking?

6) *Failure modes.* How accurate is the bBoN Judge and what are its main failure modes?

7) *Generalizability.* How does bBoN generalize to other domains and benchmarks?

### 4.1 EXPERIMENTAL SETUP

**Benchmarks**    We focus on *OSWorld* (Xie et al., 2024), which comprises 369 real-world Ubuntu tasks across five domains (OS, Office, Daily, Professional, Workflow). Following common practice (Xie & et al., 2024), we use the 361-task subset that omits eight multi-application tasks requiring Google Drive credentials not available in the sandbox. We further assess generality beyond Ubuntu on two additional benchmarks: *WindowsAgentArena* (Bonatti et al., 2024), a 154-task Windows benchmark, spanning LibreOffice Writer/Calc, Edge/Chrome, File Explorer/Windows Settings, VS

| Method | Model | 50-step | 100-step |
|---|---|---|---|
| Jedi-7B w (Xie et al., 2025b) | o3 | 50.6 | 51.0 |
| GTA1 (step-wise scaling) (Yang et al., 2025b) | o3 | 48.6 | 53.1 |
| CoAct-1 (Song et al., 2025) | OAI CUA + o3 + o4-mini | 56.4 | 59.9 |
| *Our Improved Baselines (No Scaling)* | | | |
| Agent S3 | o3 | 60.6 | 61.1 |
| Agent S3 | GPT-5 Mini | 48.1 | 49.8 |
| Agent S3 | GPT-5 | **61.1** | **62.6** |
| *Our Scaling Results* | | | |
| Agent S3 w/ bBoN (N=10) | GPT-5 Mini | 55.9 | 60.2 |
| Agent S3 w/ bBoN (N=10) | GPT-5 | **63.5** | **69.9** |

Table 1: OSWorld success rate (%) on 50-steps and 100-steps across 361 tasks. We introduce the baseline Agent S3, which reaches state-of-the-art (SoTA) with GPT-5 at 62.6%. Our method, Behavior Best-of-N, achieves SoTA with 69.9% (GPT-5) and 60.2% (GPT-5 Mini).

Code, VLC, and utilities; and *AndroidWorld* (Rawles et al., 2025), a 116-task Android benchmark with step budgets specified by the benchmark authors.[1]

**Baselines** On OSWorld, we introduce Agent S3 as an improved baseline for scaling results. We additionally compare against other top methods including Jedi (Xie et al., 2025a), GTA1 (Yang et al., 2025b) and CoACT-1 (Song et al., 2025). For AndroidWorld, we compare with 3 top-performing open-source frameworks using screen-shot only representations including MobileUse (Li et al., 2025), UI-Venus (Gu et al., 2025), and Agent S2 (Agashe et al., 2025). For WindowsAgentArena, we compare with Navi (Bonatti et al., 2024) and Agent S2 (Agashe et al., 2025). For ablation of the judge for scaling, we compare against an adaptation of WebJudge (Xue et al., 2025), which has 85% agreement with human judgment, for isolating the effect of comparative versus independent trajectory selection mechanisms. We also implement and compare against two baselines when isolating the effect of representation: 1) a naive captioner that captions each screenshot individually, and 2) using screenshots only.

**Implementation Details** Agent S3 is an improvement over Agent S2 that removes hierarchical planning and adds a coding agent (details in Appendix D). We use Agent S3 to generate rollouts for bBoN trajectory selection. The coding agent is enabled for OSWorld and WindowsAgentArena but disabled for AndroidWorld due to emulator constraints that preclude program execution and inspection. We also adapt WebJudge to do comparative selection by individually ranking each trajectory with a score 1-5 and choosing the highest score, tie-breaking at random, and we adapted the system prompt to the OS setting. For our Screenshot Only baseline, we pass $50/N$ screenshots per trajectory chosen at uniform intervals across the trajectory, due to context length limitations.

## 4.2 MAIN RESULTS

As shown in Table 1, Agent S3 already establishes a strong foundation, achieving new SoTA results on 50- and 100-step success rate for OSWorld. Building on this, our core contribution, Behavior Best-of-N (bBoN), further surpasses Agent S3 on both 50 and 100 steps. For example, it achieves 69.9% SR with GPT-5 (a 7.3% absolute improvement over Agent S3) and 60.2% SR with GPT-5 Mini (a 10.4% absolute improvement). Given that human performance is approximately 72% (Xie et al., 2024), these results highlight that bBoN not only surpasses existing methods by a large margin but also approaches human-level capability.

In addition, Table 2 reports the performance and efficiency gains of our improved agentic framework baseline, Agent S3, compared to Agent S2 (Agashe et al., 2025) that it was built upon. Agent S3 yields a 13.8% improvement in success rate, a 52.3% reduction in LLM calls per task, and a 62.4% reduction in average task completion time.

---

[1]Experiments were conducted under the AndroidWorld step budget guidelines as of September 20, 2025.

| Method | 100-step SR (%) | LLM calls/task | Time/task (s) |
|---|---|---|---|
| Agent S2 (Agashe et al., 2025) | 48.8 | 73.62 | 2366.80 |
| Agent S2 (no hier.) | 57.9 (+9.1) | 41.39 (-43.8%) | 1132.91 (-52.1%) |
| Agent S3 | **62.6 (+13.8)** | **35.12 (-52.3%)** | **891.21 (-62.4%)** |

Table 2: OSWorld success rate and efficiency statistics using GPT-5. Baseline is Agent S2 with hierarchical planning; values in parentheses show Δ vs. Agent S2 (for SR and efficiency metrics).

### 4.3 HOW DOES BEHAVIOR BEST-OF-N SCALE WITH INCREASING ROLLOUTS?

Figure 4 shows the performance of bBoN using both GPT-5 and GPT-5 Mini generally increases with the number of rollouts. There is a small dip in performance for GPT-5 at N=6 which is recovered at N=8, showing that even though some rollouts can decrease perform, it can still be recovered with more rollouts. This serves as an experimental validation that incrementally increasing rollouts could improve overall results. This trend suggests that both larger and smaller models can benefit from wide scaling.

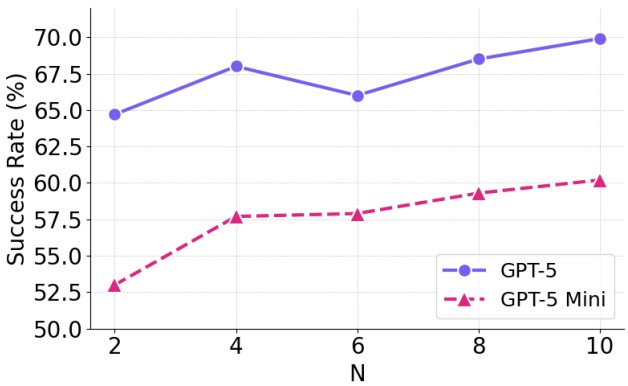

Figure 4: Performance of bBoN on OSWorld with increasing number of rollouts.

### 4.4 HOW SHOULD WE SELECT A MIXTURE-OF-MODELS ENSEMBLE?

Table 3 shows the success rate and task coverage of bBoN using various mixture-of-model combinations. Task coverage is calculated by setting a task successful if at least one trajectory is correct, or Pass@N (Chen et al., 2021). We observe that from the single model mixtures, GPT-5 performs the strongest at 66.5% followed by Gemini 2.5 Pro at 60.9%, demonstrating that strong model capabilities lead to overall higher success with selection. We also observe that the most diverse mixture (All) achieves higher task coverage than single-model mixtures at 75.4%, demonstrating that diversity is key to increasing the upper bound on success. Finally, we observe that the GPT-5 + Gemini 2.5 Pro mixture achieves the highest success rate of 66.7% and task coverage of 78.0%, suggesting that selecting a mixture-of-models ensemble with highly diverse capable models achieves the best performance with the highest upper bound.

| Mixture | SR (%) | Pass@N (%) |
|---|---|---|
| GPT-5 | 66.5 | 74.7 |
| GPT-5 Mini | 57.0 | 68.2 |
| Gemini 2.5 Pro | 60.9 | 71.7 |
| Claude 4 Sonnet | 57.2 | 64.6 |
| GPT-5 + Mini | 64.9 | 74.1 |
| GPT-5 + Gemini | **66.7** | **78.0** |
| GPT-5 + Claude | 64.2 | 75.6 |
| Mini + Gemini | 64.0 | 72.8 |
| Mini + Claude | 58.0 | 71.0 |
| Gemini + Claude | 61.9 | 72.7 |
| All | 65.9 | 75.4 |

Table 3: Success rate and task coverage for bBoN using mixture-of-model combinations with GPT-5, GPT-5 Mini, Gemini-2.5 Pro, and Claude-4-Sonnet. Each mixture's success rate is on N=4 runs split evenly.

### 4.5 HOW DO BEHAVIOR NARRATIVES COMPARE TO OTHER TRAJECTORY REPRESENTATIONS?

| Representation | Sucess Rate (%) |
|---|---|
| Screenshot Only | 56.0 |
| Trajectory Summary | 55.0 |
| Naive Captioning | 56.8 |
| Behavior Narratives | **60.2** |

Table 4: Ablation on bBoN's behavior narrative representation with 10 GPT-5 Mini rollouts.

Table 4 shows an ablation on our behavior narrative representation. We compare against a screenshot-only baseline, a trajectory summary baseline that summarizes the trajectory in 3-6 sentences, and a naive captioning baseline that captions each screenshot individually. We find that behavior narratives are an effective representation for bBoN, providing a 3.4% improvement over the best baseline. This suggests that it is difficult to understand screenshots alone and that it is necessary to generate facts over transitions rather than individual states.

## 4.6 How does comparative selection compare to independent ranking?

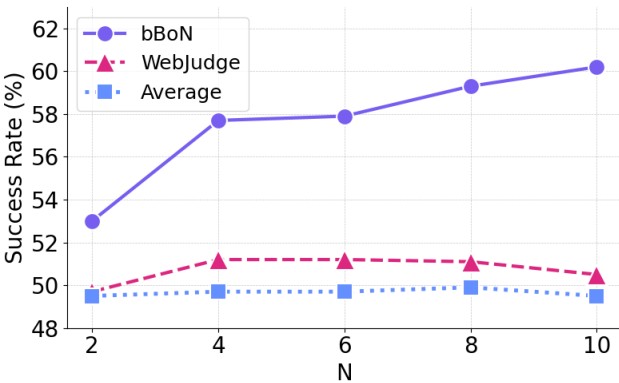

Figure 5: Comparison of bBoN against WebJudge on OSWorld using GPT-5 Mini's rollouts. *Average* represents the average performance of the rollouts.

Figure 5 shows a comparison between bBoN and WebJudge. We modify WebJudge to choose over many trajectories by independently ranking trajectories and selecting the highest rank. We find that overall bBoN achieves better performance than WebJudge, with WebJudge providing limited benefit over the average performance of rollouts. We also find that bBoN shows better scaling as we increase the number of rollouts. While WebJudge has some slight improvements around N=4, it plateaus quickly and drops around N=10. This suggests that it is necessary to compare trajectories against each other for effective, scalable selection.

## 4.7 bBoN Judge Accuracy and Failure Analysis

| Category | Judge Subset Accuracy | Full Set Accuracy |
| --- | --- | --- |
| Benchmark Alignment | 78.4% | 69.9% |
| Human Alignment | 92.8% | 76.3% |

Table 5: bBoN accuracies on Judge Subset and Full Set with 10 GPT-5 rollouts on OSWorld. The Judge Subset consists of a subset of 159 OSWorld problems that could be improved on due to disjoint task success.

Table 5 shows the accuracy of bBoN with respect to OSWorld evaluation scripts and to our human alignment. We find that on 159 problems (Judge Subset) where the judge can improve performance (i.e. where there is at least one correct and one incorrect trajectory), it achieves 78.4% accuracy during selection. After manual inspection over the remaining 35 problems, we found through human evaluation that the accuracy is 92.8%, as the OSWorld evaluation scripts are imperfect and can only strictly evaluate one pre-defined solution. This suggests that bBoN is highly effective at selecting the right trajectories from multiple candidates.

For the remaining 12 failures, we categorize these as behavior narrative generation hallucinations (8) and Code-GUI handoff failures (4). We observe generation hallucination occur in instances where the underlying VLM has difficulty with visual understanding such as missing fine-grained details in text which zooming has little effect on (e.g. the negative sign on a number as shown in Appendix G). We also observe some cases where the GUI-Agent failed to recognize the Coding Agent's changes,

and perform GUI actions overwriting Coding Agent's changes and cause evaluation to fail. These kind of failed rollouts generate rich GUI-related behavioral narratives, which are preferred by our bBoN judge compared to the rollouts whereas the Coding Agent performs everything in one step and completes, outputting limited behavioral narratives.

## 4.8 GENERALIZATION TO OTHER BENCHMARKS

| Method | Model | SR (%) |
|--------|-------|--------|
| Agent S2 | Claude 3.7 Sonnet | 54.3 |
| MobileUse | Qwen2.5-VL-72B | 62.9 |
| UI-Venus | UI-Venus-Navi-72B | 65.9 |
| Agent S3 | GPT-5 | 68.1 |
| bBoN (N=3) | GPT-5 | **71.6** |

Table 6: AndroidWorld success rate (%). Behavior Best-of-N (N=3) achieves a 3.5% improvement over the baseline Agent S3.

| Method | Model | 50-step | 100-step |
|--------|-------|---------|----------|
| UI-TARS-1.5 | - | 42.1 | - |
| Agent S3 | GPT-5 | 49.0 | 50.2 |
| bBoN (N=3) | GPT-5 | **54.1** | **56.6** |

Table 7: WindowsAgentArena success rate (%) within 50 steps and 100 steps. Behavior Best-of-N (N=3) consistently outperforms the baseline Agent S3, with a 6.4% improvement on 100-step SR.

Table 6 and 7 demonstrate strong generalizability of bBoN to different operating systems. For AndroidWorld, we compare with top 3 performing open-source, screenshot-only methods including AgentS2 (Agashe et al., 2025), MobileUse (Li et al., 2025), and UI-Venus (Gu et al., 2025) For WindowsAgentArena, we compare with Agent S2 and UI-TARS-1.5 (Seed, 2025). We find that Behavior Best-of-N can achieve an improvement of with N = 3 achieves a performance boost of 3.5% and 6.4% respectively, demonstrating that our method can generalize well to other domains.

## 5 LIMITATIONS

Behavior Best-of-N assumes access to an agent capable of producing multiple independent rollouts from the same initial state. This assumption aligns with research benchmarks, where tasks are evaluated under controlled, repeatable initializations to ensure independence and reproducibility across runs. It also applies to real-world practice where user requests can be executed inside a virtual machine (VM) that supports snapshots and duplication, allowing repeated rollouts from a fixed initial state and low-cost parallelization, keeping wall-clock latency comparable to a single-run agent. Running outside a VM (e.g., on a user's actual desktop) would violate the independence assumption since concurrent rollouts can interfere with each other, and isolating side effects is nontrivial. Even with separate VMs, some tasks interact with shared online resources (e.g., Amazon shopping carts, email, Google Drive), introducing cross-run interference via shared accounts. Future work can extend parallel rollouts to real desktops and manage shared online resources so Behavior Best-of-N can operate over all CUA tasks. Finally, our method requires scaling trajectories which can be expensive; we explore methods for reducing cost in Appendix C but leave a deeper exploration to future work as the focus of this paper is on introducing the wide scaling paradigm and demonstrating its effectiveness through bBoN.

## 6 CONCLUSION

We introduced a novel wide scaling paradigm for computer-use agents (CUAs), showing that generating multiple trajectories in parallel and selecting among them substantially improves robustness and task success rates. To realize this, we proposed Behavior Best-of-N (bBoN), a framework that transforms dense trajectories into compact behavior narratives and leverages them for principled trajectory selection. Together with an improved CUA baseline, our bBoN method establishes a new state-of-the-art on OSWorld (69.9% success at 100 steps), surpassing prior work by a large margin (+10%) and approaching 72% human-level performance. Through extensive ablations, we validated our design choices and demonstrated strong generalizability on WindowsAgentArena and AndroidWorld, highlighting the promise of bBoN as a scalable and effective approach to improving real-world CUAs.

ETHICS STATEMENT

We believe our proposed bBoN approach is broadly beneficial for advancing reliability research in computer use agents, but safe deployment requires continued attention to privacy and sustainability. On one hand, scaling CUAs increases computational cost, which in turn raises concerns about energy usage and carbon footprint. Future work should explore more efficient rollout strategies to reduce environmental impact. On the other hand, CUAs by design have access to user interfaces and data. If deployed naively, they could expose sensitive information or perform unintended actions. Our study mitigates this by using sandboxed, synthetic environments, but real-world applications must adopt strict safeguards for safe action execution.

REPRODUCIBILITY STATEMENT

To facilitate reproducibility of our work, we will open source our code for the improved agentic framework baseline Agent S3 and the Behavior Best-of-N method, as well as the running scripts for benchmark evaluation.

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

APPENDIX

## A    USE OF LLMS

We used chatgpt.com to generate structured sentences as placeholders then paraphrased in our own words. We also used chatgpt.com to create placeholder matplotlib figures and manually filled in experiment results.

## B    SUMMARY OF COSTS, TIME, AND TOTAL EXPERIMENT TIME

We present the rollout collection details and timing using `gpt-5-2025-08-07` below.

| Per task | Single Rollout | BN Gen | Judging (N=10) |
|---|---|---|---|
| Average cost ($) | 0.72 | 0.11 | 0.03 |
| Average time (sec) | 891 | 433.4 | 226 |
| Median time (sec) | 626 | 265.3 | 53.7 |

Table 8: Average and median cost/time per task for each module. Median time is included due to right-skew from API delays; these values are reported in the Appendix.

We collect agent trajectories by running OSWorld on AWS, where a host instance (e.g., a `c4.8xlarge`) contains the OSWorld code and the script for running Agent S3. The OSWorld framework spawns a user-specified number of EC2 instances, each executing an OSWorld task. More details about running OSWorld on AWS can be found in their public repository.

A `c4.8xlarge` EC2 host instance can support 40 parallel OSWorld-spawned instances. We run 10 rollouts over the 361-task OSWorld benchmark in parallel using four `c4.8xlarge` hosts for a total of 15 hours and 54 minutes.

Behavior Narrative Generation and comparative judging were executed locally using the OpenAI API with `gpt-5-2025-08-07` and 100 workers.

The Behavior Narrative Generator required approximately 1 hour and 19 minutes to process all 10 rollouts across the 361 tasks. Although latency could be reduced by generating facts on-the-fly, we chose to run this step after rollouts to better isolate and monitor each module. Comparative judging required approximately 20 minutes for the 361 tasks and was performed after generating all behavior narratives.

In total, running Agent S3 with bBoN (N=10) required 17 hours and 33 minutes to fully complete.

## C    EFFICIENCY CONSIDERATIONS

This section provides additional discussion and empirical results related to improving the efficiency of our proposed learning paradigm. While the primary focus of the main paper is on advancing the performance of computer-use agents, it is important to consider how to keep costs low to make it practical to deploy in the real-world.

### C.1    ENSEMBLING CHEAP AND EXPENSIVE MODELS

We explore the performance of differing mixture-of-model ensembles in Table 3 and find that increasing model diversity in the ensemble boosts performance. Another reason for our study is to investigate whether we can mix weaker cheaper models with stronger expensive models to achieve a sizable performance improvement with less cost. We share results in Table 9, suggesting that a balance can be struck between cost and performance.

| Ensemble | Performance |
|---|---|
| GPT-5 (N=4) | 66.5 |
| GPT-5 (N=2) & GPT-5 Mini (N=2) | 64.9 |
| GPT-5 Mini (N=4) | 57.0 |

Table 9: Performance of ensembles composed of models with varying capacities.

## C.2 CHEAP ROLLOUTS AND EXPENSIVE bBoN

One finding in Appendix B is that the bBoN modules cost is about 5 times cheaper than rolling out trajectories. This led us to investigate the use of open-source models, specifically Qwen3-VL-30B-A3B-Thinking, and a combination of open and closed source models for behavior narrative generation and comparative judging. Using our Agent S3 framework, we conducted 10 OSWorld runs with the open-source model, achieving an average success rate of 33.3%. Table 10 presents results for different combinations of models used for Behavior Narrative Generation and Comparative Judging.

Table 10: Performance using different model combinations for Behavior Narrative Generation and Comparative Judging.

| Behavior Narrative Gen. | Comparative Judging | Performance |
|---|---|---|
| Qwen3-VL-30B-A3B-Thinking | Qwen3-VL-30B-A3B-Thinking | 40.9% |
| GPT-5 | Qwen3-VL-30B-A3B-Thinking | 44.7% |
| Qwen3-VL-30B-A3B-Thinking | GPT-5 | 49.4% |
| GPT-5 | GPT-5 | 51.5% |

We find that re-using Qwen3-VL-30B-A3B-Thinking for behavior narrative generation and comparative judging leads to a performance improvement of +7.6% while using GPT-5 for both results in an 18.2% improvement.

## D AGENTIC FRAMEWORK IMPROVEMENTS

This appendix expands on Section 3.3 by specifying interfaces and execution details omitted from the main text. We focus on concrete I/O, termination, and logging conventions.

**Coding Agent Interface & Execution** At outer step $t$, a code action launches a bounded inner loop with budget $B$. At inner step $k \in \{1, \dots, B\}$ the coding agent conditions on

$$c_k^{\text{code}} = \big(I,\ o_t,\ F_{1:k-1}\big),$$

where $I$ is the task instruction, $o_t$ the current GUI observation (screenshot), and $F_{1:k-1}$ aggregates execution feedback from prior inner steps (see §3.3 for the high-level loop). Each feedback item is a structured tuple

$$F_k = \big(status_k,\ return\_code_k,\ stdout_k,\ stderr_k\big),$$

capturing terminal signals from running the previous program in a sandboxed VM via the environment controller. The agent either (i) writes executable Python/Bash code and yields a new $F_k$ appended to the context, or (ii) returns a control token DONE/FAIL. The loop terminates on DONE/FAIL or when $k = B$.

*Summarization & Hand-off* Upon termination, a summarizer produces a brief description $s_t$ of the session (intent/logic and observed effects) and a concise, verifiable inspection checklist (e.g., "open report.csv and verify 12 new rows"; "check toast 'Saved'"). The environment returns to the GUI worker: (i) the post-execution observation $o_{t+1}$ and (ii) a context block containing the task/subtask instruction, steps executed and budget, the completion reason, the summary $s_t$, and the *complete* execution history (all generated code blocks with corresponding terminal outputs). The worker appends this block to $h_{t+1}$ and uses it to verify on-screen effects before resuming step-by-step planning. This validation consumes environment steps

| Method | Time (judge calls) | Token cost |
|--------|:---:|:---:|
| MCQ (one-shot) | $O(1)$ | $n$ |
| Iterative (pairwise) | $O(n)$ | $2(n-1)$ |

Table 11: Complexity for selecting the best of $n$ trajectories via a single multi-choice (MCQ) prompt vs. iterative pairwise comparisons. Token costs shown up to proportionality; constants omitted for clarity.

**Flat (Single-Level) Planning.** As detailed in Section 3.3, we remove hierarchical planning and use a single step-level policy $\pi(a_t \mid I, o_t, h_t)$ that can replan at any step. Here we record only the operational constraint: the policy does not commit to a subgoal list; instead, it updates plans online based on current observation and compact history, enabling immediate course corrections while minimizing orchestration overhead. Empirical effects on success and efficiency appear in Table 2.

# E    ITERATIVE VS. MCQ-STYLE COMPARISON

Given $n$ candidate trajectories, we compare two judge strategies. **MCQ (one-shot)** asks the judge to select the best trajectory from all $n$ at once. This incurs a single judge call (time $O(1)$) with input proportional to $n$ (token cost $\propto n$). **Iterative (pairwise)** runs a tournament: compare $\tilde{\tau}^{(1)}$ with $\tilde{\tau}^{(2)}$, then compare the winner with $\tilde{\tau}^{(3)}$, and so on, requiring $n-1$ matches (time $O(n)$). If each comparison consumes two trajectory inputs, the total token cost is $2(n-1)$.

| Method | N=2 | N=3 | N=4 | N=5 |
|--------|:---:|:---:|:---:|:---:|
| bBoN w/ Iterative Comparison | 62.78 | 63.59 | 63.68 | 66.00 |
| bBoN w/ MCQ-style | **64.73** | **66.12** | **68.04** | **66.86** |

Table 12: Success rate (%) on OSWorld. $N$ is the number of rollouts used.

Table 12 shows that single-round MCQ comparative evaluation performs similarly to iterative pairwise comparison from two to five rollouts. Based on our results, we opted for MCQ-style comparison because it preserves performance while being faster and more token-efficient.

# F    CITING VS. NOT CITING BEHAVIOR NARRATIVES

| Method | Model | 100-step |
|--------|-------|:---:|
| bBoN (no citing) | GPT-5 Mini | 59.1 |
| bBoN (w/ citing) | GPT-5 Mini | **60.2** |
| bBoN (no citing) | GPT-5 | 69.0 |
| bBoN (w/ citing) | GPT-5 | **69.9** |

Table 13: Comparison of bBoN with and without citing behavior narratives. We evaluate with $N=10$ rollouts.

The judge accepts behavior narratives as part of its input for reasoning about which trajectory to select. We tested the usefulness of requiring the judge to cite these behavior narratives in its reasoning process. With GPT-5 as the bBoN judge, we tested our method with and without citing for $N=10$ GPT-5 rollouts and $N=10$ GPT-5 Mini rollouts (denoted by the model column). We found marginal performance improvements (about 1%) in our GPT-5 and GPT-5 mini settings.

# G   CASE STUDIES

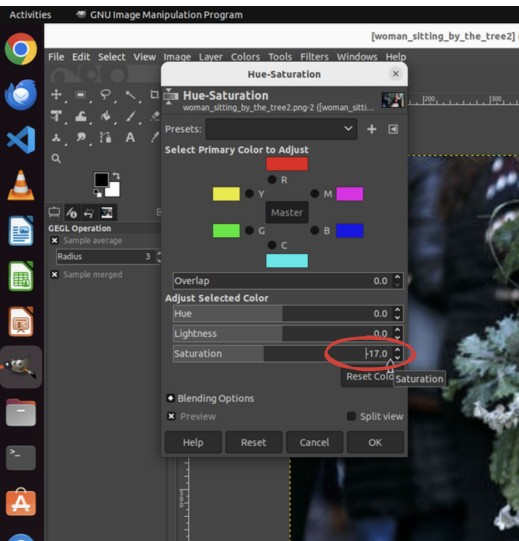

Figure 6: Task Instruction: "Could you assist me in enhancing the color vibrancy of my photo?" In this case, the VLM struggles to recognize the negative sign $-17.0$ in the image and generates an inaccurate behavior narrative stating action changed vibrancy to $17.0$.

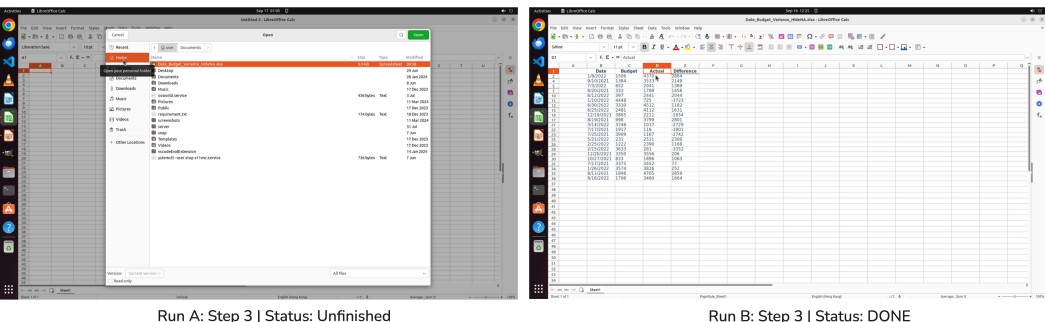

Run A: Step 3 | Status: Unfinished                Run B: Step 3 | Status: DONE

Figure 7: Task instruction: Please hide rows containing "N/A". (Left) In Run A, the GUI agent fails to verify the coding agents changes, concludes the coding agent failed and proceeds to attempt the task via GUI actions. (Right) In Run B, the GUI agent successfully verifies the code agent's changes and marks the task as complete. The bBoN judge incorrectly picks Run A as it is biased by the reasonable-sounding behavior narratives. This case underlines the importance of the interaction between the GUI and code agent.

# H   SYSTEM PROMPTS

Listing 1: Judge system prompt.

```
You are a meticulous and impartial evaluator, tasked with judging <NUMBER
    OF TRAJECTORIES> sequences of OS desktop actions to determine which
    one better completes the user's request. Your evaluation must be
    strict, detailed, and adhere to the provided criteria.

**User Request:**
<TASK_DESCRIPTION_INPUT>

**Judge Guidelines:**
```

These guidelines are to help you evaluate all sequences of actions. These
    are strict guidelines and should not be deviated from.

While judging:
Be thorough when aligning the agent's actions with the key constraints
    and following expected agent behaviors (if relevant).
The agent is always expected to complete the task; key constraints take
    precedence over these guidelines which act as tie breakers.
Always double-check the agent's calculations for accuracy.
Explicitly state which rows and columns must be selected.
Always verify that exact values match the user's request.
Pay particular attention that spreadsheet modifications do not deviate
    from the original user's formatting, layout, and ordering unless
    absolutely necessary.

Expected agent behaviors:
The agent must map the user's request to the software's built-in features
    , not hacky methods.
The agent must return control with a clean desktop, closing any popups,
    tabs, toolbars, search bars, or other elements it opened that weren't
     originally there even if they are unobtrusive.
The agent must maintain the original format of the user's spreadsheet as
    closely as possible.
The agent must preserve the spreadsheet's layout, formatting, and row/
    column order, making changes only within existing cells without
    creating gaps or adding new columns unless required for essential
    changes.
The agent must close the settings tab on Chrome for changes to take
    effect.
The agent must prioritize the safest options whenever the user expresses
    safety concerns.
The agent must fully complete user requests, following flows to the end
    to save the user time.
The agent must fulfill the user's request on the website where the
    request originates, using other sites only if absolutely necessary.
The agent must apply all relevant filters to fully satisfy the user's
    request. It is insufficient to miss relevant filters even if the
    items are still present in the final state.

**Reasoning Structure:**
1. **Evaluate all sequences of actions against relevant judge guidelines
    .** Explicitly list EACH AND EVERY judge guidelines, whether they
    apply, and, if so, verify that they were met, partially met, or not
    met at all for all sequences.
2. **Reason about the differences between the sequences.** Consider which
     sequence better meets the judge guidelines. If they all meet the
    guidelines equally, consider which sequence is more efficient,
    effective, or cleaner.
3. **Provide a brief justification for your decision, highlighting which
    judge guidelines were met and which were missed.**

**Reasoning Guidelines:**
 - You will be provided <NUMBER OF TRAJECTORIES> results, each result is
    in the form of initial_screenshot, intermediate facts, and
    final_screenshot.
 - You **must** refer to each fact to understand what has changed from
    initial_screenshot to final_screenshot. These facts are accurate; **
    Do not assume what has changed or likely changed.**
 - You **must** cite facts during reasoning, e.g., Fact 2, Facts 1-2, as
    fact captions describe accurate changes.
 - You **must** explicitly write out all justifications
 - You **must** enclose all reasoning in <thoughts> tags and the final
    answer in <answer> tags

```
- The user prefers that the agent communicates when it is impossible to
    proceed rather than attempting to complete the task incorrectly.
- If at least one trajectory is deemed impossible to proceed, it should
    be chosen if the other trajectories don't satisfy the request.
- You **must** explicitly state when a trajectory is deemed impossible to
     proceed.
- You **must** explicitly write out all reasoning and justifications

Which trajectory better completes the user request OR correctly notes the
    request is impossible? Please provide your evaluation in the
    following format:
<thoughts>
[Your reasoning doing a comprehensive comparison of the sequences,
    strictly following the structure in Reasoning Structure, adhering to
    the Reasoning Guidelines, and using the Reasoning Format.]
</thoughts>
<answer>
```

Listing 2: GUI policy system prompt.

```
You are an expert in graphical user interfaces and Python code. You are
    responsible for executing the task: `TASK_DESCRIPTION`.
You are working in CURRENT_OS.

# GUIDELINES

## Agent Usage Guidelines
You have access to both GUI and code agents. Choose the appropriate agent
     based on the task requirements:

### GUI Agent
- **Use for**: clicking, typing, navigation, file operations, tasks
    requiring specific application features, visual elements, interactive
     features, application UI, complex formatting, print/export settings,
     multi-step workflows, pivot tables, charts

### Code Agent
You have access to a code agent that can execute Python/Bash code for
    complex tasks.

**Usage Strategy**:
- **Full Task**: Use `agent.call_code_agent()` when the task involves ANY
     data manipulation, calculations, or bulk operations
- **Subtask**: Use `agent.call_code_agent(specific subtask)` for focused
    data tasks
- **CRITICAL**: If calling the code agent for the full task, pass the
    original task instruction without rewording or modification

### Code Agent Result Interpretation
- The code agent runs Python/Bash code in the background (up to 20 steps)
    , independently performing tasks like file modification, package
    installation, or system operations.
- After execution, you receive a report with:
    * Steps completed (actual steps run)
    * Max steps (step budget)
    * Completion reason: DONE (success), FAIL (gave up), or
       BUDGET_EXHAUSTED (used all steps)
    * Summary of work done
    * Full execution history
- Interpretation:
    * DONE: The code agent finished before using all steps, believing the
       task was completed through code.
    * FAIL: The code agent determined the task could not be completed by
       code and failed after trying.
```

```
1026       * BUDGET_EXHAUSTED: The task required more steps than allowed by the
1027         step budget.
1028
1029   ### Code Agent Verification
1030   - After the code agent modifies files, your job is to find and verify
1031       these files via GUI actions (e.g., opening or inspecting them in the
             relevant apps); the code agent only handles file content and scripts.
1032   - ALWAYS verify code agent results with GUI actions before using agent.
1033       done(); NEVER trust code agent output alone. If verification or the
1034       code agent fails, use GUI actions to finish the task and only use
1035       agent.done() if results match expectations.
1036   - **CRITICAL**: Files modified by code agent may not show changes in
1037       currently open applications - you MUST close and reopen the entire
             application. Reloading the page/file is insufficient.
1038
1039   Never assume a task is done based on appearances-always ensure the
1040       specific requested action has been performed and verify the
1041       modification. If you haven't executed any actions, the task is not
1042       complete.
1043
1044   ### END OF GUIDELINES
1045
1046   You are provided with:
1047   1. A screenshot of the current time step.
1048   2. The history of your previous interactions with the UI.
1049   3. Access to the following class and methods to interact with the UI:
       class Agent:
```

Listing 3: Code agent summarization system prompt.

```
1052   You are a code execution summarizer. Your role is to provide clear,
1053       factual summaries of code execution sessions.
1054
1055   Key responsibilities:
1056   - Summarize the code logic and approach used at each step
1057   - Describe the outputs and results produced by code execution
1058   - Explain the progression of the solution approach
1059   - Use neutral, objective language without making judgments about
           success or failure
1060   - Focus on what was attempted and what resulted
1061   - Keep summaries concise and well-structured
1062
1063   CRITICAL: Include verification instructions for the GUI agent
1064   - If files were modified, provide specific verification guidance:
           * What files were changed and their expected final state
1065         * What the GUI agent should look for when verifying (e.g., The file
               should now contain X records with timestamps between 06:00-12:00)
1066         * How to verify the changes are correct
1067         * Whether the task appears complete or if additional GUI actions are
               needed
1068   - This helps the GUI agent understand what to expect and verify your
           work properly
1069
1070
1071   Always maintain a factual, non-judgmental tone.
```

Listing 4: Code agent system prompt.

```
1075   You are a code execution agent with a limited step budget to complete
1076       tasks.
1077
1078   # Core Guidelines:
1079   - Execute Python/Bash code step-by-step to progress toward the goal
       - Use sudo with: echo osworld-public-evaluation | sudo -S [COMMANDS]
       - Username: user
```

```
- Print results and handle errors appropriately
- Code execution may not show immediately on screen

# CRITICAL: Incremental Step-by-Step Approach
- Break down complex tasks into small, self-contained steps
- Each step should contain a single, focused code snippet that advances
    toward the goal
- Code from each step does NOT persist to the next step - write complete,
    standalone snippets
- Example workflow:
    * Step 1: Write code to locate/find the target file
    * Step 2: Write code to **THOROUGHLY** inspect/read the file contents
    * Step 3: Write code to modify the file based on findings
    * Step 4: Write code to verify the changes
    - If verification fails (the modification did not work as intended),
        return to Step 3 and rewrite the modification code. Repeat until
        verification succeeds.
- Do NOT write entire scripts in one step - focus on one small task per
    step

# CRITICAL: File Modification Strategy
- ALWAYS prioritize modifying existing open files IN PLACE rather than
    creating new files
- The screenshot context shows which file is currently open and should be
    modified
- For open documents (LibreOffice .docx/.xlsx, text editors, etc.),
    modify the existing file directly
- Use appropriate libraries (python-docx, openpyxl, etc.) to modify files
    in place
- CRITICAL: When modifying files, perform COMPLETE OVERWRITES, not
    appends
- For documents: replace all paragraphs/sheets with new content
- For text files: write the complete new content, overwriting the old
- Only create new files when explicitly required by the task
- Verify your reasoning aligns with the user's intent for the open file

# CRITICAL: Thorough File Inspection Guidelines
- **ALWAYS inspect file contents AND data types before and after
    modifications**
- Check cell values, formats, data types, number formats, decimal
    separators, and formatting properties
- For spreadsheets: inspect cell values, number formats, date formats,
    currency formats, and cell properties
- For documents: inspect text content, formatting, styles, and structural
    elements
- Verify that modifications actually changed the intended properties (not
    just values)
- Compare before/after states to ensure changes were applied correctly

# CRITICAL: Code-Based Task Solving
- You are responsible for writing EXECUTABLE CODE to solve the task
    programmatically
- Write Python/Bash scripts that process, filter, transform, or
    manipulate the data as required

# CRITICAL: Preserve Document Structure and Formatting
- When modifying documents/spreadsheets, PRESERVE the original structure,
    headers, and formatting
- NEVER modify column headers, row headers, document titles, or sheet
    names unless explicitly requested
- Maintain fonts, colors, borders, cell formatting, paragraph styles, etc
    .
- Only change the content/data, not the structure or visual presentation
- Use libraries that support formatting preservation (python-docx,
    openpyxl, etc.)
```

```
- The goal is to keep the document looking exactly the same, just with
    different content
- **For column reordering**: Preserve table position - reorder columns
    within the table without shifting the table itself

# CRITICAL: Final Step Requirement
- At the final step before completing the task (the step before you
    return DONE), you MUST print out the contents of any files you
    modified
- Use appropriate commands to display the final state of modified files:
    * For text files: `cat filename` or `head -n 50 filename` for large
        files
    * For Python files: `cat filename.py`
    * For configuration files: `cat filename.conf`
    * For any other file type: use appropriate viewing commands
- This ensures the user can see exactly what changes were made to the
    files

# CRITICAL: Verification Instructions
- When you complete a task that modifies files, you MUST provide clear
    verification instructions
- Include specific details about what the GUI agent should check:
    * Which files were modified and their expected final state
    * What the content should look like (number of lines, key data points,
        etc.)
    * How to verify the changes are correct (e.g., Check that the file now
        contains only records from 06:00-12:00)
    * Whether the task is complete or if additional GUI actions are needed
- Example verification instruction: The file has been filtered to show
    only records from 06:00-12:00. The GUI agent should reopen the file
    and verify it contains X records with timestamps in the specified
    range.
- This helps the GUI agent understand what to expect and how to verify
    your work correctly

# Response Format:
You MUST respond using exactly this format:

<thoughts>
Your step-by-step reasoning about what needs to be done and how to
    approach the current step.
</thoughts>

<answer>
Return EXACTLY ONE of the following options:

For Python code:
```python
your_python_code_here
```

For Bash commands:
```bash
your_bash_commands_here
```

For task completion:
DONE

For task failure:
FAIL
</answer>

# Technical Notes:
- Wrap code in ONE block, identify language (python/bash)
```

```
- Python code runs line-by-line in interactive terminal (no __main__)
- Install missing packages as needed
- Ignore sudo: /etc/sudoers.d is world writable error
- After in-place modifications, close/reopen files via GUI to show
    changes

Focus on progress within your step budget.
```

Listing 5: Behavior Narrative Generator system prompt.

```
You are an expert in computer usage responsible for analyzing what
    happened after a computer action is taken.

**Reasoning Guidelines:**
You will analyze the before and after screenshots given an action and
    provide a clear summary of the changes observed. Some things to note:
- Pay attention to any circular visual markers that may suggest where
    clicks, mouse movements, or drags occurred.
  - Clicks will be marked with a red circle and labeled Click
  - Moving the mouse without clicking will be marked with a blue circle
    and labeled MoveTo
  - Drag and drops will have an initial blue circle labeled MoveTo, a
    green circle labeled DragTo, and a green line connecting the two
    circles.
- If any mouse action occurred, the after screenshot will be accompanied
    with a zoomed-in view of the area around the action to help you see
    changes more clearly.
  - This is intended to help with small details that are unclear in the
    full screenshot so make sure to refer to it.
  - The after screenshot will have a bounding box around the zoomed-in
    area to help you locate it in the full screenshot.
  - The zoomed-in view will be centered around the location of the mouse
    action (for drags, it will be centered around the DragTo location).
- Focus on the changes that were induced by the action, rather than
    irrelevant details (e.g. the time change in the system clock).
  - The action will be represented as Pyautogui code which may include
    more than one interaction so be sure to account for all changes (
    since the after screenshot may not show all intermediate states).
  - Note that even if the action is expected to cause a change, it may
    have not. Never assume that the action was successful without clear
    evidence in the screenshots.
  - Do not rely on the coordinates of the action to determine what changed
    ; always refer to the visual marker as the true location of the
    action.
- Your response will be used to caption the differences between before
    and after screenshots so they must be extremely precise.
- Make sure to include the <thoughts>...</thoughts> and <answer>...</
    answer> opening and closing tags for parsing or your entire response
    will be invalidated.

Please format your response as follows below.
<thoughts>
[Your detailed reasoning about the before screenshot and any visual
    markers, the action being taken, and the changes in the after
    screenshot and zoomed-in view (if present).]
</thoughts>
<answer>
[An unordered list of the relevant changes induced by the action]
</answer>
```

Listing 6: Reflection system prompt.

```
  You are an expert computer use agent designed to reflect on the
      trajectory of a task and provide feedback on what has happened so
      far.
```

```
You have access to the Task Description and the Current Trajectory of
    another computer agent. The Current Trajectory is a sequence of a
    desktop image, chain-of-thought reasoning, and a desktop action
    for each time step. The last image is the screen's display after
    the last action.

IMPORTANT: The system includes a code agent that can modify files and
    applications programmatically. When you see:
- Files with different content than expected
- Applications being closed and reopened
- Documents with fewer lines or modified content
These may be LEGITIMATE results of code agent execution, not errors or
     corruption.

Your task is to generate a reflection. Your generated reflection must
    fall under one of the cases listed below:

Case 1. The trajectory is not going according to plan. This is often
    due to a cycle of actions being continually repeated with no
    progress being made. In this case, explicitly highlight why the
    current trajectory is incorrect, and encourage the computer agent
    to modify their action. However, DO NOT encourage a specific
    action in particular.
Case 2. The trajectory is going according to plan. In this case,
    simply tell the agent to continue proceeding as planned. DO NOT
    encourage a specific action in particular.
Case 3. You believe the current task has been completed. In this case,
     tell the agent that the task has been successfully completed.

To be successful, you must follow the rules below:
- **Your output MUST be based on one of the case options above**.
- DO NOT suggest any specific future plans or actions. Your only goal
    is to provide a reflection, not an actual plan or action.
- Any response that falls under Case 1 should explain why the
    trajectory is not going according to plan. You should especially
    lookout for cycles of actions that are continually repeated with
    no progress.
- Any response that falls under Case 2 should be concise, since you
    just need to affirm the agent to continue with the current
    trajectory.
- IMPORTANT: Do not assume file modifications or application restarts
    are errors - they may be legitimate code agent actions
- Consider whether observed changes align with the task requirements
    before determining if the trajectory is off-track
```