# OpenReview forum: "The Unreasonable Effectiveness of Scaling Agents for Computer Use"
_ICLR.cc/2026/Conference — Submitted to ICLR 2026_

### Official Review · Reviewer_77YS · 2025-10-21

**Soundness:** 2
**Presentation:** 3
**Contribution:** 2
**Rating:** 4
**Confidence:** 3

**Summary:**

The paper introduces bBoN, a “wide‑scaling” paradigm for computer‑use agents that generates multiple trajectories in parallel and selects the best via narrative comparison, a concept well‑framed and clearly presented in the abstract. However, it provides little discussion of the substantial computational cost such scaling entails.

**Strengths:**

The paper tackles a central issue in computer‑use agents—how to scale behavioral selection effectively—which is timely and relevant as larger models are increasingly limited by interaction efficiency.

The proposed bBoN framework is clearly formulated, with intuitive motivation and a well‑explained pipeline from trajectory generation to narrative‑based comparison.

The experiments, especially the comparison in Figure 5, provide concrete evidence that bBoN outperforms both average‑rollout and simple WebJudge baselines, illustrating the benefit of cross‑trajectory evaluation.

**Weaknesses:**

Although bBoN achieves better performance through multiple rollouts and narrative evaluations, this improvement comes with a potentially huge computational overhead. The paper does not quantitatively analyze cost, efficiency, or fairness under equal‑budget conditions, leaving readers uncertain about the real‑world practicality of the method.

The main contribution—running many rollouts in parallel and selecting via pairwise narrative evaluation—follows a relatively straightforward ensemble logic: performance generally improves when sampling more trajectories and filtering them with a learned judge. While well‑executed, this approach feels incremental rather than conceptually innovative, relying on existing ideas of scaling and comparative evaluation.

The method relies on aggregating multiple rollouts into a single narrative summary for comparison. As the number of rollouts or the length of each trajectory increases, both token‑level and computational costs grow rapidly. Moreover, very long context windows may dilute key information and amplify model attention errors, leading to less reliable judgments. The paper does not analyze how accuracy or reasoning quality scales with context length, which raises concerns about robustness in large‑scale or long‑horizon tasks.

**Questions:**

How does the framework handle potential side effects or environment resets when multiple rollouts are executed repeatedly? Do many resets cost more?

Does the narrative summarization stage require multimodal input (e.g., multiple images per rollout)? If so, how is this handled given that some current LLM/VLM APIs only support single‑image inputs?

What is the additional inference or token cost introduced by the narrative‑comparison stage?

In Section 3.2, the paper describes the Behavior Best‑of‑N Judge as performing a multiple‑choice (MCQ) comparative evaluation over N candidate trajectories. However, the corresponding system prompt (Appendix) only supports pair‑wise comparison (“which one better completes the user request”), without any structure for handling N > 2 options. This discrepancy suggests that the actual implementation is Best‑of‑2 rather than the claimed Best‑of‑N？

---

> ### Author Response · Authors · 2025-11-18
>
> We thank the reviewer for their comments for our paper and for recognizing the importance of our work, the clear description of the method, and the clear evidence that comparative judging is effective over single-rollout judges like WebJudge. We address your concerns below.
>
> > The paper does not quantitatively analyze cost, efficiency, or fairness under equal‑budget conditions, leaving readers uncertain about the real‑world practicality of the method.
>
> We re-emphasize that the focus of our paper is on scaling rollouts to approach human-level performance; it would be premature to investigate the efficiency of our wide scaling paradigm without sizable performance gains. Since our method closes the gap by a large margin, we believe improving the efficiency of the wide-scaling paradigm is a great next step for future work. We refer the reviewer to the global response section for our discussion on efficiency as well as summary on costs and time.
>
> > While well‑executed, this approach feels incremental rather than conceptually innovative, relying on existing ideas of scaling and comparative evaluation.
>
> We humbly disagree with the suggestion that this work is incremental. In Section 2.2, we acknowledge that test-time scaling has been applied to CUA in a step-wise BoN manner; however, our work, bBoN, is the first to scale at the trajectory-level and implement trajectory-level comparative evaluation. bBoN also demonstrates impressive, near-human-level performance: 69.9% vs. 72% human-level on OSWorld, being the first work to reach this close.
>
> Applying BoN for trajectories in CUA is difficult because, in the worst case, it would require analyzing upwards of fifty to a hundred images of the environment state which introduces context limit issues and trajectory understanding complications. Our solution is using behavior narratives as a representation that makes it tractable to perform trajectory-wise BoN over long-horizon, multimodal trajectories.
>
> > The method relies on aggregating multiple rollouts into a single narrative summary for comparison.
>
> We believe the reviewer misunderstands our method and wish to clarify that our method does not aggregate multiple rollouts into a single narrative summary. Instead, we compress one rollout into one narrative, and use multiple narratives for comparison.
>
> We would like to restate our method to clarify the confusion:
> Each rollout is converted into a Behavior Narrative through the Behavior Narrative Generator. This module works by converting each transition, represented by the before screenshot, action, and after screenshot, into a fact. The initial screenshot, accompanied by the sequence of generated facts, and the final screenshot represents the Behavior Narrative which describes what the agent did for a single trajectory. Finally, our judge compares these N rollouts through analyzing their respective Behavior Narratives.
>
> > How does the framework handle potential side effects or environment resets when multiple rollouts are executed repeatedly? Do many resets cost more?
>
> Rollouts are executed in parallel and in separate VMs. Each task is self-contained inside a VM and the VM’s state is reset after the agent completes the task; there are no environment resets to reuse VMs. Regarding side effects, the only side effects of this framework are in tasks that have a shared server-state which we address in the limitations section. Future directions may build stronger infrastructure, allowing for parallel runs using shared server states.
>
> > Does the narrative summarization stage require multimodal input (e.g., multiple images per rollout)? If so, how is this handled given that some current LLM/VLM APIs only support single‑image inputs?
>
> To clarify, our rollouts for a given task are converted to Behavior Narratives through the Behavior Narrative Generator. This module’s purpose is to capture, or summarize, the changes within the environment state after an action has been taken. This module requires the before and after image of the environment for an action. The output of this module, a Behavior Narrative, for a rollout is multimodal, consisting of the initial screenshot, the facts generated, and the final screenshot.
>
> Our bBoN Judge is also multimodal as it compares these Behavior Narratives to select the correct trajectory. We are unfamiliar with any VLM APIs that only support single-image inputs. Our experiments utilize the OpenAI, Anthropic, and Google Gemini APIs which are all multimodal and support multiple images.
>
> > What is the additional inference or token cost introduced by the narrative‑comparison stage?
>
> The average cost of the comparative judging is $0.03 utilizing gpt-5-2025-08-07. We refer the reviewer to the global response section on the summary of cost for further details.

---

> > ### Author Response · Authors · 2025-11-18
> >
> > > In Section 3.2, the paper describes the Behavior Best‑of‑N Judge as performing a multiple‑choice (MCQ) comparative evaluation over N candidate trajectories. However, the corresponding system prompt (Appendix) only supports pair‑wise comparison (“which one better completes the user request”), without any structure for handling N > 2 options. This discrepancy suggests that the actual implementation is Best‑of‑2 rather than the claimed Best‑of‑N?
> >
> > The actual implementation is the claimed Best-of-N. The system prompt takes in a <NUMBER_OF_TRAJECTORIES> variable to inform the judge of how many trajectories to compare. Some of the wording in the Appendix prompt may include artifacts from the iterative pairwise approach we initially tried (which we found to consistently underperform compared to MCQ-style approach; results in Appendix C), but the input to the judge are the N rollouts’ Behavior Narratives and we’ve qualitatively verified that the judge reasons about all rollouts.
> >
> > Below is a snippet (due to character limits) of our revised prompt which has been included in the paper revision.
> >
> > ```python
> > judge_system_prompt = """
> > You are a meticulous and impartial evaluator, tasked with judging <NUMBER OF TRAJECTORIES> sequences of OS desktop actions to determine which one better completes the user's request. Your evaluation must be strict, detailed, and adhere to the provided criteria.
> >
> >
> > [... omitted for space...]
> >
> >
> > **Reasoning Structure:**
> > 1. **Evaluate all sequences of actions against relevant judge guidelines.** Explicitly list EACH AND EVERY judge guidelines, whether they apply, and, if so, verify that they were met, partially met, or not met at all for all sequences.
> > 2. **Reason about the differences between the sequences.** Consider which sequence better meets the judge guidelines. If they all meet the guidelines equally, consider which sequence is more efficient, effective, or cleaner.
> > 3. **Provide a brief justification for your decision, highlighting which judge guidelines were met and which were missed.**
> >
> > **Reasoning Guidelines:**
> >   - You will be provided <NUMBER OF TRAJECTORIES> results, each result is in the form of initial_screenshot, intermediate facts, and final_screenshot.
> >   - You **must** refer to each fact to understand what has changed from initial_screenshot to final_screenshot. These facts are accurate; **Do not assume what has changed or likely changed.**
> >   - You **must** cite facts during reasoning, e.g., Fact 2, Facts 1-2, as fact captions describe accurate changes.
> >   - You **must** explicitly write out all justifications
> >   - You **must** enclose all reasoning in <thoughts> tags and the final answer in <answer> tags
> >
> > - The user prefers that the agent communicates when it is impossible to proceed rather than attempting to complete the task incorrectly.
> > - If at least one trajectory is deemed impossible to proceed, it should be chosen if the other trajectories don't satisfy the request.
> > - You **must** explicitly state when a trajectory is deemed impossible to proceed.
> > - You **must** explicitly write out all reasoning and justifications
> >
> > Which trajectory better completes the user request OR correctly notes the request is impossible? Please provide your evaluation in the following format:
> > <thoughts>
> > [Your reasoning doing a comprehensive comparison of the sequences, strictly following the structure in Reasoning Structure, adhering to the Reasoning Guidelines, and using the Reasoning Format.]
> > </thoughts>
> > <answer>
> > [The index of the better sequence, a single integer from 1 to <NUMBER OF TRAJECTORIES>]
> > </answer>"""
> > ```

---

> > > ### Comment · Reviewer_77YS · 2025-11-27
> > >
> > > The rebuttal usefully clarifies the pipeline (one Behavior Narrative per rollout, separate VMs with reset, and a Best-of-
> > > N judge), so my earlier concerns about implementation details are largely resolved.
> > >
> > > My main concerns remain. First, there is still no end-to-end or equal-budget cost analysis: beyond a rough per-call cost for the judge, we do not see total token/compute usage (including multi-image rollouts and narrative generation), scaling curves as N and trajectory length grow, or fixed-budget comparisons to strong single-rollout or lighter test-time scaling baselines. Since it is well-known that Best-of-N can beat Best-of-1 when resources are unconstrained, this makes the real-world practicality hard to evaluate.
> > >
> > > Second, even after rebuttal, the method still reads as a well-engineered instantiation of “sample many trajectories and let a strong judge pick the best,” i.e., an ensembling/scaling pattern, rather than a fundamentally new algorithmic idea, despite the solid Behavior Narrative design and strong OSWorld results.
> > >
> > > Third, while my earlier misunderstanding about “aggregating” rollouts was corrected, the judge still operates over multiple long multimodal narratives, and there is no analysis of how judgment quality scales with context length or larger N.

---

> ### Author Response · Authors · 2025-12-03
>
> Thank you for the response, we are glad we could clarify our pipeline for you. We address your other concerns below which we have incorporated in the paper revision.
>
> # Regarding end-to-end cost analysis
>
> In our initial response where we responded with the “per-call cost for the judge” we additionally referred the reviewer to the global response with the remaining per-task cost details for rollouts and behavior narrative generation. We include the table below and
>
> | Per task                 | Single Rollout | BN Gen   | Judging (N=10) |
> |--------------------------|--------------------|---------------|----------------------|
> | Average cost         |      $0.72         |  $0.11       |          $0.03        |
> | Total cost               |      $259.62    |   $39.70     |         $10.83      |
>
> Overall our total experiment cost for our best result using gpt-5-2025-08-07 for N=10 rollouts over 100 steps on OSWorld (69.9% in Table 1) was $3004.03 or about $8.32 for a single task. The rollouts account for 86.4% of the cost which we believe should be where future work should focus on optimizing to lower the cost of our method.
>
> # Regarding equal-budget cost analysis
>
> We will address this concern by first re-iterating our focus in this work and then describing why an equal-budget cost analysis is out of scope.
>
> Our focus is to introduce a new paradigm, wide-scaling via bBoN, which enables test-time scaling to work for computer-use agents. The prevailing method before our work was GTA-1 [1] which performs test-time scaling by comparing next action candidates. Although helpful for local improvements, this commits agents to the initial agent plan which could be incorrect or make it harder to execute other plans. Thus, our work investigates wide-scaling where full trajectory candidates are compared.
>
> But performing Best-of-N selection over trajectory candidates is challenging because trajectories are information-dense and multimodal, where many details are irrelevant to task success and relevant details may be difficult to understand. Thus, representing trajectories and distinguishing the correct trajectory from the rest is not trivial. To address this challenge, we compared various trajectory representations in Section 4.5 as well as how a judge should distinguish between trajectories in Section 4.6 and Appendix D.
>
> We believe that an equal-budget cost analysis between a “strong single-rollout” and our test-time scaling approach with bBoN would not be meaningful. Consider an equal-step budget: we can compare a single rollout with a step budget of 1000 steps to 10 rollouts of 100 steps using bBoN; however, this would be misleading as, in practice, the single rollout would terminate earlier, leaving much of its budget unused. If we alleviate this by running an equal-token budget to the strong single-rollout, we would expect our approach to drop to near-zero accuracy. Assuming that tokens and steps are proportional, equal-token comparison of a single-rollout and 2 rollouts would halve the number of steps to complete the task, making it impossible to achieve success.
>
> Instead our paper includes “scaling curves as N grow” over increasing values of N (2,4,6,8,10) in Sections 4.3 and 4.6 and “as trajectory length grow” over increasing steps in Table 1 and 7. We have also include GTA1’s results in Table 1 as a “lighter test-time scaling baseline”. We acknowledge the concern of efficiency but emphasize we do expect that “resources are unconstrained”, simply that with increasing resources we found evidence of increasing gains using our approach compared to other judge approaches that naively aggregate (Section 4.6). To make this more explicit, we have added new content to Limitations and in the Appendix that summarizes our global response which shares our initial efforts into making our approach more efficient and which directions we believe are most fruitful for future work to tackle.
>
> [1] Yang et. al 2025. GTA1: GUI Test-time Scaling Agent.

---

> ### Author Response · Authors · 2025-12-03
>
> # Regarding our contributions
>
> We do not claim to be proposing a “fundamentally new algorithmic idea” as ensembling and scaling through Best-of-N selection is a common practice, but it is not trivial how to apply this practice in multimodal, multi-turn computer-use environments. Some applications include BoN on step-level actions [1] or aggregating LLM-as-a-judge scores across various trajectories (Section 4.6), but we show that both of these approaches do not achieve significant gains. In contrast, our approach Behavior Best-of-N performs BoN over a behavior narrative representation that compresses the relevant information in a trajectory.
>
> We restate our contributions as follows: (1) we introduce the wide scaling paradigm for CUA where selecting over multiple trajectories enables robustness and coverage (2) we propose Behavior Best-of-N as a framework that effectively instantiates this paradigm (3) we achieve SoTA (69.9%) on OSWorld with our strong CUA baseline, Agent S3, combined with Behavior Best-of-N and (4) we provide ablations that clearly demonstrate the necessity of our design choices and results that show strong zero-shot generalizability on other domains.
>
> # Regarding judgement quality
>
> We use performance as a proxy for analyzing judgement quality which we did investigate with scaling curves of increasing values of N (2,4,6,8,10) in Sections 4.3 and 4.6 and with increasing context length via larger trajectories in Table 1 and 7. Since the performance generally improves as the number of rollouts increases, we conclude that the judgement quality remains strong for higher N.

---

### Official Review · Reviewer_q8SD · 2025-10-29

**Soundness:** 3
**Presentation:** 3
**Contribution:** 2
**Rating:** 2
**Confidence:** 4

**Summary:**

The paper introduces a test-time scaling framework for computer-use agents called Behavior Best-of-N (bBoN): it runs multiple full task trajectories in parallel and then selects the best one at the end, greatly improving reliability on long, brittle workflows. To make different candidates directly comparable, it compresses each step into a concise Behavior Narrative that records the action and resulting screen change, and uses a multi-way bBoN judge to pick the strongest trajectory rather than scoring runs in isolation. The authors also streamline the underlying agent (Agent S3) by removing heavy hierarchical planning and integrating a coding agent, so it can generate stronger candidates faster. Together, these components yield higher success rates and better cross-platform generalization (e.g., to Windows and Android) under practical step budgets.

**Strengths:**

- Running multiple full trajectories in parallel and selecting the best (bBoN) reduces single-run brittleness in long-horizon tasks and provides predictable gains as the number of candidates increases.
- Behavior Narratives distill each step into factual “action → screen change” summaries, enabling a multiway judge to compare candidates directly, more reliable and scalable than independent scoring or pairwise elimination.
- The streamlined base agent (Agent S3) produces stronger rollouts with lower overhead, and the full stack shows consistent success-rate improvements and positive cross-platform transfer (e.g., Windows/Android) under realistic step budgets.

**Weaknesses:**

- Missing implementation details for parallel rollouts：The paper claims Best-of-N by running multiple full trajectories concurrently but omits a reproducible execution recipe: no specification of multi-machine vs. single-host isolation, environment cloning/reset procedures (images/snapshots/templates), seeding and cross-run isolation (caches, network, account state), or resource-parity policies (CPU/GPU/IO). This gap undermines reproducibility and comparability.
- Best-of-N increases cost roughly with N, yet the paper does not report wall-clock time, CPU/GPU hours, energy, or cost-normalized success. There is no equal-compute baseline comparison (e.g., single/multi-sample methods under the same total budget), so observed gains may primarily reflect more sampling rather than a more efficient method.
- Gains may hinge on heterogeneous candidate pools (different backbones, prompts, temperatures). Without isolating diversity vs. pure sampling, it’s unclear whether bBoN helps when all candidates come from the same model/config.
- The method is demonstrated on tasks with clear success signals. It remains unclear how the judge handles open-ended goals, partial credit, or ambiguous outcomes where “best” is not binary.
- The work does not characterize marginal gains as N grows or provide guidance to select N under a fixed budget, making it hard to tune for real-time constraints.

**Questions:**

- When generating N parallel trajectories, what infrastructure do you use (multi-machine vs. single host with multiple VMs/containers), how are environments cloned/reset, and how are seeds and resource quotas set to ensure isolation and fairness across candidates? Could you release scripts/configurations to reproduce the same parallel conditions?
- Under equal total compute budgets, how do bBoN results compare to single-run or few-sample baselines, and what wall-clock/CPU-GPU hours and energy are reported?
- Do gains depend on heterogeneous candidate pools (different models/prompts/temperatures), or do they persist when all candidates come from the same model and config?
- Are there early-exit policies to stop clearly bad candidates mid-run, or adaptive N strategies by task difficulty to control latency and cost at test time?
- How does bBoN perform under UI/layout drift, app updates, ads/pop-ups, and network jitter, particularly when initial snapshots cannot perfectly normalize the environment?

---

> ### Author Response · Authors · 2025-11-18
>
> We thank the reviewer for their insight into our paper and for recognizing the reliability, scalability, and generalizability of our approach. We address your concerns below.
>
> > Missing implementation details for parallel rollouts
>
> OSWorld is multi-machine in that it uses a Host-Client architecture where the host instance (the controller) launches and manages multiple client instances (VMs running OSWorld tasks). Each VM dynamically created by OSWorld for a task is isolated from other VMs. These VMs are all created from the same, fixed AMI snapshot. Given each VM starts from the same initial state, each task has a preconfig which initializes the task (e.g. downloading files or opening apps to prepare the environment for task execution). These VMs are deleted automatically at the end of the task and the AMI remains unchanged. The instructions for setup and running are included in the [PUBLIC_EVALUATION_GUIDELINE.md](https://github.com/xlang-ai/OSWorld/blob/main/PUBLIC_EVALUATION_GUIDELINE.md) in their public repository.
>
> > …the paper does not report wall-clock time
>
> Actually, we do report wall-clock time in Section 4.2 Main Results. Table 2 shows the latency speed-up improvements from Agent S2 to Agent S3. We refer the reviewer to the global response on the summary of time for average and median times for behavior narrative generation and comparative judging.
>
> > There is no equal-compute baseline comparison… Are there early-exit policies to stop clearly bad candidates mid-run, or adaptive N strategies by task difficulty to control latency and cost at test time?
>
> An equal-compute comparison is not meaningful in the context of our paper because our baselines are single-rollout methods while our bBoN method selects actions by evaluating multiple rollouts. These approaches utilize compute in fundamentally different ways. To force an “equal compute” comparison, we would need to extend the single-rollout baseline to run for 100×N steps to compare against N rollouts; however, we observe that agents typically terminate before the 100-step limit (only 1.6% of the 10 gpt-5-2025-08-07 rollouts exceed steps) making this comparison uninformative.
>
> We re-emphasize that the focus of our paper is on scaling rollouts to approach human-level performance which demands increasing compute. Since our method closes the human-level gap by a large margin, we believe improving the efficiency of the wide-scaling paradigm is a great next step for future work. We refer the reviewer to the global response section for our discussion on efficiency.
>
> Regarding the reviewer’s suggestions, early-exit policies is an efficiency strategy that can now be enabled through our new paradigm. In OSWorld, a FAIL action is used for infeasible tasks that the agent should recognize as impossible; however, we observe the agent commonly calls this FAIL action when it believes no progress has been made after a substantial amount of tries and plans. This is detrimental for single rollout runs since the performance-optimal strategy would be to rollout the maximal number of steps before calling the DONE action. In our paradigm, we can encourage this behavior to prune poor runs early as other runs may complete the task more efficiently.
>
> Regarding adaptive N strategies, this would require estimating task difficulty. One simple heuristic can be binning into 5 difficulty categories based on the model’s pass@1 [1] on a curated training set per each domain. This allows us to select N=2,4,6,8,10 based on the difficulty of the domain. This disregards differing task difficulty within a domain but allows us to avoid incurring additional compute to estimate task difficulty prior to performing a task.
>
> > Gains may hinge on heterogeneous candidate pools (different backbones, prompts, temperatures). Without isolating diversity vs. pure sampling, it’s unclear whether bBoN helps when all candidates come from the same model/config.
>
> This appears to be a misunderstanding from the reviewer. We clearly specify that our main results in Table 2 utilize multiple rollouts from the same model, improving our score from 62.6 for gpt-5-2025-08-07 for N=1 to 69.9 when N=10. We have an experiment dedicated to studying heterogeneous candidate pools in Section 4.4. Our experiments show that using gpt-5-2025-08-07 and gemini-2.5-pro achieves higher performance and Pass@4, suggesting a mixture-of-models ensemble with diverse, capable models is important for achieving the best performance.
>
> [1] Scaling LLM Test-Time Compute Optimally can be More Effective than Scaling Model Parameters. ICLR 2025

---

> > ### Author Response · Authors · 2025-11-18
> >
> > > How does bBoN perform under UI/layout drift, app updates, ads/pop-ups, and network jitter, particularly when initial snapshots cannot perfectly normalize the environment?
> >
> > OSWorld tasks are run using an Ubuntu VM which naturally contains stochasticity (e.g. pop-up ads on Chrome, varying download times due to network and continually updated websites).  We qualitatively observe bBoN shows a strong ability to select successful trajectories that overcome these stochastic challenges. Fundamentally, bBoN allows parallel runs of the same task and selects the best run among them, which bypasses the noise and variance of one single run. Therefore, it is much more robust and stable than single-run baselines.

---

### Official Review · Reviewer_sHwe · 2025-10-31

**Soundness:** 3
**Presentation:** 3
**Contribution:** 3
**Rating:** 6
**Confidence:** 3

**Summary:**

This work proposes a method for improving the performance of computer-use agents by generating multiple rollouts (wide scaling), summarizing their behavior by generating facts per action using a VLM (Behavior Narrative Generator), and selecting the best one through comparative evaluation using a VLM (Behavior Best-of-N Judge). The proposed framework achieves state-of-the-art performance on OSWorld, a standard benchmark for computer-use agents, in success rate for long-horizon tasks.

**Strengths:**

- Experimental setup: Detailed experimental evaluation, including ablation studies, baselines, and datasets.

- Strong empirical performance: Achieves state-of-the-art success rate on OSWorld.

- Clear presentation: The method is well-described and motivated.

**Weaknesses:**

- Inefficiency: Naive wide scaling means running N times more rollouts, with no mechanism to prune unpromising rollouts early or summarize fewer behaviors.

- No learning or adaptation: The method relies on the performance of pre-trained LLMs and VLMs in a zero-shot setting.

- VLM dependency: The approach depends on pre-trained VLMs both for summarizing behavior and for selecting the best trajectory, which may compound errors.

**Questions:**

- Could you add in Table 2 the runtime for different BoN values? Is it expected to be exactly N× that of Agent S3?

- How many steps does an optimal agent typically need to solve OSWorld tasks? Why did you choose 50 and 100 steps? Could you report the statistics (e.g., mean, median, variance) for the number of steps of an optimal agent?

- What is the impact of visual augmentations for pointer interactions on the performance of the Behavior Narrative Generator? Could you provide further details on the visual augmentation method?

-  Have you tried generating facts for the entire behavior with a single prompt (i.e., a trajectory-level summary) instead of per-action summaries?

---

> ### Author Response · Authors · 2025-11-18
>
> We thank the reviewer for their response and are glad they found the experimental setup and methods section detailed and clear. We respond to your questions below.
>
> > Could you add in Table 2 the runtime for different BoN values? Is it expected to be exactly N× that of Agent S3?
>
> The expected runtime is much lower than Nx that of Agent S3 because we run rollouts in parallel.  An average single Agent S3 run is about 5 hours and 18 minutes with 40 parallel VMs, each for a task. Running 10 rollouts sequentially would take 53 hours. Running 10 parallel rollouts would achieve 1x the time of a single Agent S3 run. However, due to resource constraints we ran 4 parallel rollouts (each running 40 tasks in parallel) amounting to about 15 hours and 54 minutes. We refer the reviewer to the global response on the summary of costs, time and total experiment time for more information.
>
> > How many steps does an optimal agent typically need to solve OSWorld tasks? …Could you report the statistics (e.g., mean, median, variance) for the number of steps of an optimal agent?
>
> It is intractable to calculate the number of steps an optimal agent needs to solve a task due; just considering the click action on a 1920x1080 screen results in a branching factor of nearly 2 million. Some works [1] attempt to compare agent efficiency against human trajectories considered as “near-optimal” but the action spaces aren’t directly comparable (e.g. an agent can be “beyond optimal” through code execution that requires humans multiple steps). Instead, we provide the statistics for the number of steps on our best gpt-5-2025-08-07 run from [Agent S3 w/ bBoN (N=10)]. Our best run averaged 15.8 steps per task or a median of 9.2 steps per task. The standard deviation is 18.6 steps as certain domains (e.g. multi-apps) generally take more steps.
>
> > Why did you choose 50 and 100 steps?
>
> Our primary focus is studying long-horizon trajectories since small mistakes that CUAs make can propagate and lead to failure, making them unreliable. We chose the 50 and 100 steps settings because the performance of prior state-of-the-art methods and competitive baselines were achieved using 50 and 100 steps, which are considered long-horizon for current agents. We did not evaluate with a max step limit above 100 because we found the agent rarely reaches the 100 step limit (1.6% of trajectories for all 10 rollouts of gpt-5-2025-08-07).
>
> > What is the impact of visual augmentations for pointer interactions on the performance of the Behavior Narrative Generator?
>
> When developing the Behavior Narrative Generator, our priority was to ensure that the generated facts were of high quality. Towards this end, we added (1) click annotations to the before screenshot to make it clearer where an action would occur, and (2) a bounding box and zoom crop of the affected area to clearly show the effect of the action.
>
> We test bBoN without visual augmentations with the same setting as our best run Agent S3 with bBoN N=10 (69.9% as reported in Table 1) and observed bBoN achieves 68.1% on gpt-5-2025-08-07 rollouts without visual augmentation, compared to the 69.9% reported in Table 1. Our results indicate that including visual augmentations leads to a near 2% boost on overall performance, demonstrating that our visual augmentations aid the judge in making more accurate decisions.
>
> > Could you provide further details on the visual augmentation method?
>
> We refer the reviewer to Section 3.1 for a description of our visual augmentations. Our Behavior Narrative Generator receives as input a before screenshot, the action taken by the agent, and the after screenshot of the environment state. This module annotates the location of the coordinate action on the before screenshot with a red dot and text about the action type. The after screenshot is annotated with a bounding box around the location of the coordinate action. These 2 annotated screenshots, the agent action, and a zoomed-in area of the after screenshot where the bounding box is are fed into the Behavior Narrative Generator which generates a description of what environmental change was induced by the agent action.
>
> [1] OSWorld-Human: Benchmarking the Efficiency of Computer-Use Agents

---

> ### Author Response · Authors · 2025-11-18
>
> > Have you tried generating facts for the entire behavior with a single prompt (i.e., a trajectory-level summary) instead of per-action summaries?
>
> We provide our results for generating entire behavior narratives with a single trajectory-level summary below which we have included in the paper revision. Our trajectory-level summary is a single prompt instructing the VLM to summarize in 3-6 sentences what the agent attempted, key UI transitions and actions, and the final outcome (whether the task succeeded or failed). The VLM receives the full agent action history and thinking traces along with as many screenshots of the environment state as can fit into context (following our Screenshot Only baseline protocol under Implementation Details in Section 4.1).
>
> We tested this with gpt-5-2025-08-07 rollouts and compared with our gpt-5-2025-08-07 results in Figure 4. Our results indicate trajectory-level summarization consistently underperforms bBoN. This may be due to the LLM’s inability to reason thoroughly across a long sequence of screenshots and actions. An information-dense trajectory of screenshots and actions may be difficult for the LLM to reason about because less emphasis is given on the individual actions and state changes.
>
> |                      | N=2  | N=4  | N=6  | N=8  | N=10  |
> |----------------------|------|------|------|------|-------|
> | bBoN (from Figure 4) | 64.7 | 68.0 | 66.0 | 68.5 | 69.9  |
> | Traj. Summary        | 64.1 | 65.4 | 65.2 | 66.3 | 65.7  |

---

> > ### Comment · Reviewer_sHwe · 2025-11-25
> > **Response to Authors**
> >
> > Thank you for their detailed responses to my questions and for running the additional trajectory-level experiment. The mentioned weaknesses regarding inefficiency, lack of learning/adaptation, and VLM dependence are inherent to the proposed approach and remain unaddressed. Due to this, I will maintain my recommendation of a weak accept, while acknowledging that the proposed method achieves strong empirical results.

---

> > > ### Author Response · Authors · 2025-11-26
> > >
> > > Thank you for the response! We address the reviewer’s concerns regarding efficiency, learning and adaptation, and VLM dependence below.
> > >
> > > # Regarding efficiency
> > > We would like to reiterate that, though efficiency is crucial, our paper’s primary aim is to scale rollouts to approach human-level performance on computer-use tasks. Now that our method significantly narrows the gap, as the reviewer has pointed out by “strong empirical results”, we believe improving the efficiency of our framework is a promising direction for future research. We refer the reviewer to the global response section for our discussion on efficiency and cost and time summaries.
> > >
> > > Regarding the mechanism for pruning unpromising rollouts early, this is one strategy that is enabled through our new paradigm. In OSWorld, a FAIL action exists for infeasible tasks that the agent should recognize are impossible. We’ve commonly observed that the agent decides to call this action when it believes no significant progress is being made. This is a detrimental strategy for single rollouts since the performance-optimal strategy would be to utilize the maximal number of steps before calling the DONE action. In our paradigm, we believe that we can reinforce this behavior to prune poor runs earlier as there is a chance for success in other rollouts.
> > >
> > > Regarding summarizing fewer behaviors, we believe the reviewer means generating less facts per behavior narratives. We find in our global response that the average cost of generating the behavior narrative for a single trajectory is roughly 6.5x cheaper than rolling out the trajectory. Given this cost gap, we think that improving rollout efficiency through pruning may offer a more impactful direction; however there are also ways to make these modules more efficient such as generating behavior narrative facts in larger batches than step-level facts (i.e. summarizing intermediate trajectories based off key milestones).
> > >
> > > # Regarding learning and adaptation
> > >
> > > We believe there is nothing wrong with relying on the performance of existing pre-trained LLMs or VLMs for test-time scaling as other ICLR acceptances have relied on in the past [1,2]. The focus of our paper is on test-time scaling over a representation that improves long-horizon understanding using existing model capabilities, not necessarily training the model for this capability. We found that existing model capabilities were sufficient for significant performance gains with our bBoN approach. In fact, we claim it is a strength that our method works zero-shot with existing model capabilities as it can be seamlessly integrated with any existing agent to boost performance.
> > >
> > > # Regarding VLM dependency
> > >
> > > We acknowledge that our modular system for generating behavior narratives, and judging trajectories using behavior narratives could compound errors; however, we do not find this to be a major concern based on our human alignment investigation in Section 4.7 which we summarize below.
> > >
> > > | Category            | Judge Subset Accuracy | Full Set Accuracy |
> > > |---------------------|-----------------------|-------------------|
> > > | Benchmark Alignment |                 78.4% |             69.9% |
> > > | Human Alignment     |                 92.8% | 76.3%             |
> > >
> > > Our results show that the bBoN judge achieves 92.8% alignment with human evaluation on the 159 OSWorld problems where the judge meaningfully influences performance (the Judge Subset). This represents a 14.4-point improvement over the benchmark’s alignment with human judgment and a 6.4-point gain in overall OSWorld performance. These findings indicate that even with VLM-generated narratives and judging, the selected rollouts remain well aligned with human evaluation.
> > >
> > > [1] Wang, Xuezhi, et al. "Self-consistency improves chain of thought reasoning in language models." ICLR 2023
> > >
> > > [2] Snell, Charlie, et al. "Scaling llm test-time compute optimally can be more effective than scaling model parameters." ICLR 2025 Oral.

---

### Official Review · Reviewer_muQu · 2025-11-01

**Soundness:** 1
**Presentation:** 2
**Contribution:** 2
**Rating:** 4
**Confidence:** 3

**Summary:**

This paper proposes a Behavior Best-of-N (bBoN) method, which scales over agents by generating multiple rollouts and selecting the best trajectory among them for execution.

**Strengths:**

The paper explores the scaling over agents by a simple but effective best-of-N method, which largely improves the performance in benchmarks.

The paper proposes an effective Behavior Judge method that could support the scaling, which is better than the WebJudge.

**Weaknesses:**

1.  The proposed method is not practical. It requires a simulator or virtual machine to perform multiple rollouts before real execution, which limits its applicability and can incur excessive computational costs during testing, especially when the N is set to a large number.


2. This is also based on a strong assumption that the environment's transition dynamics are not static and that the simulator could be exactly the same as the real environment. This is difficult to guarantee in the real world, as websites are not stationary and can always have pop-ups or ads [1]. Thus, although the proposed method can achieve better performance in benchmarks, it is actually overfitting to the benchmark, which cannot represent real-world performance.

3. As one can access the simulator, it would be better to run the baseline the same N times and use the task success indicator to select the successful trial among the N trials. This allows for calculating the success rate for each task, which can serve as the performance upper bound of the best-of-N method.


[1] DigiRL: Training In-The-Wild Device-Control Agents with Autonomous Reinforcement Learning.

**Questions:**

See the weakness.

---

> ### Author Response · Authors · 2025-11-18
>
> We thank the reviewer for their comments. We are glad the reviewer acknowledges the large improvements across benchmarks and the effectiveness of the Behavior Narrative judge. We address your concerns below.
>
> > The proposed method is not practical. It requires a simulator or virtual machine to perform multiple rollouts before real execution
>
> Our method is practical to do using virtual machines (VM) and it is common industry practice to run agents within VMs such as OpenAI Operator [1] and Google Project Mariner [2].
>
> Our method is also practical to apply to desktops with strong infrastructure. We strongly believe that agents can work in parallel through infrastructure that allows performing browser tasks on separate tabs [2] or duplicating documents to make concurrent edits.
>
> Our paper scope assumes that rollouts can be run independently to simplify our new wide-scaling paradigm and serve as a proof of concept to encourage others to explore it further in practical applications or future research such as using bBoN to select high-quality rollouts for training within a simulator.
>
> > The proposed method… can incur excessive computational costs during testing
>
> We re-emphasize that the focus of our paper is on scaling rollouts to approach human-level performance which inevitably incurs computational cost. Since our method closes the gap to human-level performance on OSWorld by a significant margin, we believe improving the efficiency of the wide-scaling paradigm is a great next step for future work. We refer the reviewer to the global response section for our discussion on efficiency.
>
> > This is also based on a strong assumption that the environment's transition dynamics are not static and that the simulator could be exactly the same as the real environment. This is difficult to guarantee in the real world, as websites are not stationary and can always have pop-ups or ads…although the proposed method can achieve better performance in benchmarks, it is actually overfitting to the benchmark
>
> There seems to be a misunderstanding from the reviewer. The simulators in the benchmarks we utilize are virtual machines of real-world operating systems and are not simplified sandboxes. OSWorld uses Ubuntu VMs, WindowsAgentArena uses Windows VMs, and AndroidWorld uses an Android emulator. Moreover, OSWorld and WindowsAgentArena include browser tasks with actual real-world webpages that present the challenge of stochasticity in different parallel runs, such as UI updates, pop-up ads, captchas, and varying loading speeds.
>
> We are not overfitting to OSWorld as 1) stochasticity exists in OSWorld tasks (e.g. browser tasks) and 2) we provide experiments in Section 4.8 on AndroidWorld (Android) and WindowsAgentArena (Windows) to share that our framework can generalize to operating systems other than Ubuntu on OSWorld, where the majority of our experiments are focused.
>
> > As one can access the simulator, it would be better to run the baseline the same N times and use the task success indicator to select the successful trial among the N trials.
>
> This is exactly why our bBoN method is practical for the real world tasks, as it does not need the task success indicator (aka, ground truth task evaluator). Access to a simulator does not necessarily mean a task success indicator is available. For the vast majority of real world tasks beyond benchmarks, there are no task success indicators to do the selection. Therefore, our bBoN method uses a VLM-based judge to select among rollouts instead of relying on the unrealistic assumption of having access to a task success indicator.
>
> [1] https://openai.com/index/computer-using-agent/
>
> [2] https://deepmind.google/models/project-mariner/

---

### Author Response · Authors · 2025-11-18
**Global Response**

We thank all the reviewers for their feedback for our submission. In this section we respond to common concerns raised by the reviewers.

# Summary of costs, time and total experiment time

Below we summarize the average cost and time of each module per task. We include the median time as API delays caused our average to be heavily right-skewed. This has been included in the paper revision Appendix.

| Per task                 | Single Rollout | BN Gen   | Judging (N=10) |
|--------------------------|--------------------|---------------|-----------------------|
| Average cost         |      $0.72         |  $0.11        |          $0.03        |
| Average time (sec)|        891          |  433.4       |            226         |
| Median time (sec) |        626           |   265.3      |           53.7         |

We present the rollout collection details and time using gpt-5-2025-08-07 as our model below.

We collect our agent trajectories by running OSWorld on AWS where a host instance (e.g. c4.8xlarge) contains the OSWorld code and our script to run Agent S3. The OSWorld code spawns a user-specified number of EC2 instances, each of which runs an OSWorld task. More details for how to run OSWorld on AWS can be found in their public repository.

A c4.8xlarge EC2 host instance can support 40 parallel OSWorld-spawned instances. We run 10 rollouts over the OSWorld benchmark (361 tasks) in parallel using 4 c4.8xlarge instances for a total of 15 hours and 54 minutes.

We ran the Behavior Narrative Generation and comparative judging locally with OpenAI API  gpt-5-2025-08-07 using 100 workers.

The Behavior Narrative Generator took about 1 hour and 19 minutes for all 10 rollouts on each of the 361 tasks. To further optimize for latency, we could have generated facts on-the-fly, but decided to run them after rollouts to individually monitor each module’s runs. Comparative judging took about 20 minutes for all 361 tasks and was run after generating every behavior narrative.

In total, running Agent S3 w/ bBoN (N=10) took 17 hours and 33 minutes to fully complete.

# Concerns about efficiency

We agree it is important to ensure the efficiency of our approach given we scale over many rollouts; however, our focus in this paper is mainly on improving the performance of computer-use agents which are still lagging behind human-level performance. We achieve SoTA by a large margin demonstrating the effectiveness of our new learning paradigm, narrowing the gap between human and Compute-Use Agent from about 10% (Agent S3’s average performance) to 2%. We believe exploring such techniques in future work would be appropriate to improve the efficiency of our new paradigm.

## Ensemble of cheap and expensive models maintains higher performance

One experiment we’ve included in the paper under Section 4.4 explores performance with mixture-of-models. We find that increasing model diversity in the ensemble boosts performance. Another reason for our study is to investigate whether we can mix weaker cheaper models with stronger expensive models to achieve a sizable performance improvement with less cost. We share results below, suggesting that a balance can be struck between cost and performance.

| Ensemble                       | Performance |
|--------------------------------|-------------|
| GPT-5 (N=4)                    | 66.5        |
| GPT-5 (N=2) & GPT-5 Mini (N=2) | 64.9        |
| GPT-5 Mini (N=4)               | 57.0        |

## Cheap open-source rollouts and closed-source bBoN leads to large gains

After our submission, we also explored using an open-source model like Qwen3-VL-30B-A3B-Thinking. We were interested in investigating how we can utilize cheap open-source rollouts and a combination of open-source and closed-source models in the bBoN modules which are considerably cheaper (5x) than rolling out. We completed 10 OSWorld runs with this model using our Agent S3 framework, averaging 33.3% success rate. Below we present the results on the open-source model and gpt-5-2025-08-07 during Behavior Narrative generation and comparative judging.

| Behavior Narrative Generation  | Comparative Judging       | Performance |
|---------------------------|---------------------------|-------------|
| Qwen3-VL-30B-A3B-Thinking | Qwen3-VL-30B-A3B-Thinking | 40.9%       |
| GPT-5                     | Qwen3-VL-30B-A3B-Thinking | 44.7%       |
| Qwen3-VL-30B-A3B-Thinking | GPT-5                     | 49.4%       |
| GPT-5                     | GPT-5                     | 51.5%       |

We find that re-using Qwen3-VL-30B-A3B-Thinking for behavior narrative generation and comparative judging leads to a performance improvement of +7.6% while using GPT-5 for both results in an 18.2% improvement.

---

### Author Response · Authors · 2025-12-03
**Summary for AC**

We thank the reviewers for their time and effort in reviewing our paper and concisely summarize the major points below.

- Regarding real-world practicality (muQu, 77YS)
  - Running multiple rollouts from the same initial state is practical. Cloud platforms provide fast VM/container snapshotting and cloning, allowing us to regenerate identical starting states and run many rollouts in parallel. This workflow is already standard in industry [1,2] so our assumptions align with existing practice.
  - Our method could also be practical to apply to desktops with strong infrastructure. We strongly believe that agents can work in parallel through infrastructure that allows performing browser tasks on separate tabs [2] or duplicating documents to make concurrent edits.
- Regarding algorithmic novelty (77YS)
  - Our method is composed of the Behavior Narrative Generator and Comparative Best-of-N Judge which together enable wide scaling over various trajectories
  - Unlike past works [3,4,5,6,7] that judge directly off raw screenshots, we create a behavior narrative representation that extracts relevant details from information-dense screenshots to judge effectively; we demonstrate the effectiveness of our representation against other trajectory representations in Section 4.5.
  - Unlike past works [4,5,6,7] that judge single trajectories, we introduce a comparative Best-of-N judge that chooses the best trajectory by comparing multiple at once; we demonstrate the effectiveness of our judge against prior works by comparing against aggregating a single trajectory judge’s scores in Section 4.6.
- Regarding efficiency (sHwe, q8SD, 77YS)
  - The focus of our paper is to introduce a new test-time scaling paradigm, wide scaling by generating multiple trajectories and choosing the best, and investigate how to build a framework to enable this paradigm to work
  - The reviewers requested an equal-compute budget comparison between multi-trajectory approaches and single agent baselines approaches which would not be meaningful. Any test-time scaling approach would look worse in an equal-compute budget comparison. The most appropriate efficiency comparison is between bBoN and WebJudge in Section 4.6 where we show that bBoN is consistently more performant by up to 10 percentage-points.
  - Nonetheless, we have added sections to Appendix B and C where we’ve reported the average costs and added preliminary experiments about explorations into reducing the costs of rollouts, the most expensive part of our pipeline, to incentivize future work into improving the cost of our method.

And we summarize our revisions to the paper (in blue) below:
- We address Reviewer sHwe’s concern about a trajectory summary representation by adding an experiment in Section 4.5; a trajectory summary achieves 55.0% while Behavior Narrative achieves 60.2%, maintaining our claim of the effectiveness of our representation.
- We address Reviewers muQu, q8SD and 77YS concerns about missing cost and time by adding a table with the average costs and average/median time to Appendix B. We additionally add a description of our AWS VM details along with the total runtime of our experiments.
- Despite this paper’s focus on investigating a judge framework to enable the wide scaling paradigm, we address Reviewer muQu, q8SD and 77YS concerns regarding the cost of our method by including preliminary experiments in Appendix C on reducing rollout costs, the most expensive part of the pipeline, to discuss directions for future work
- We address Reviewer sHwe’s concern about how our comparative judge works (MCQ-style instead of Iterative) by updating the system prompt in Appendix H to clarify our judge chooses the best trajectory by comparing all trajectories at once.

We believe we have addressed all the reviewer’s concerns and clarified misunderstandings regarding efficiency. Our method is practical for real-world deployment via VMs and could be deployed on user desktops via parallel agent infrastructure. Our method enables and demonstrates the effectiveness of the wide scaling paradigm, as acknowledged by all reviewers through our strong empirical results. Finally, our method achieves large performance gains against comparable test-time scaling baselines in equal-compute settings. We plan to clarify the writing to make it clearer which baselines should be compared in terms of efficiency.

[1] https://openai.com/index/computer-using-agent/

[2] https://deepmind.google/models/project-mariner/

[3] Yang et. al 2025. GTA1: GUI Test-time Scaling Agent.

[4] He et. al 2024. Webvoyager: Building an end-to-end web agent with large multimodal models

[5] Deng et. al 2023. Mind2web: Towards a generalist agent for the web

[6] Xue et. al 2025. An illusion of progress? assessing the current state of web agents

[7] Gou et. al 2025. Mind2Web 2: Evaluating Agentic Search with Agent-as-a-Judge

---

### Meta-Review · Area_Chair_CqQG · 2026-01-04

**Summary:**

The reviewers' concerns are mainly about:

1. Cost/efficiency analysis and real-world practicality under constraints.

2.  “straightforward ensemble scaling” vs algorithmic contribution.

3. Robustness + scaling behavior: how performance/judgment holds as N and trajectory length increase.

I think this paper is straightforward idea with BoN. This paper is with marginal technical contributions and the results are not that surprising. The above concerns should be addressed carefully and the rebuttal is not very convincing for reviewers to accept this paper.

**Reviewer Concerns:**

I think the main remaining concerns are mainly about the technical contributions of this paper and the cost/efficiency of this methods.

**Reviewer Scores:**

I do not think the reviewers will change the scores to accept this paper.

---

### Decision · Program_Chairs · 2026-01-26

Reject